systems biology/computational biology

system biology, dynamical model, gene regulatory network, lineage commitment, landscape

**Author for correspondence:**
Mengyao Wang
e-mail: w_mengyao@shu.edu.cn

# The complex landscape of haematopoietic lineage commitments is encoded in the coarse-grained endogenous network

Mengyao Wang[1,2], Junqiang Wang[3], Xingxing Zhang[2] and Ruoshi Yuan[4]

[1]School of Life Science, and [2]Shanghai Center for Quantitative Life Sciences and Physics Department, Shanghai University, Shanghai 200444, People's Republic of China
[3]Key Laboratory of Systems Biomedicine, Shanghai Center for Systems Biomedicine, Shanghai Jiao Tong University, Shanghai 200240, People's Republic of China
[4]California Institute for Quantitative Biosciences (QB3), University of California, Berkeley, CA 94706, USA

MW, 0000-0001-7190-331X; RY, 0000-0002-9508-7326

Haematopoietic lineage commitments are presented by a canonical roadmap in which haematopoietic stem cells or multipotent progenitors (MPPs) bifurcate into progenitors of more restricted lineages and ultimately mature to terminally differentiated cells. Although transcription factors playing significant roles in cell-fate commitments have been extensively studied, integrating such knowledge into the dynamic models to understand the underlying biological mechanism remains challenging. The hypothesis and modelling approach of the endogenous network has been developed previously and tested in various biological processes and is used in the present study of haematopoietic lineage commitments. The endogenous network is constructed based on the key transcription factors and their interactions that determine haematopoietic cell-fate decisions at each lineage branchpoint. We demonstrate that the process of haematopoietic lineage commitments can be reproduced from the landscape which orchestrates robust states of network dynamics and their transitions. Furthermore, some non-trivial characteristics are unveiled in the dynamical model. Our model also predicted previously under-represented regulatory interactions and heterogeneous MPP states by which distinct differentiation routes are intermediated. Moreover, network perturbations resulting in state transitions indicate the effects of ectopic gene

expression on cellular reprogrammes. This study provides a predictive model to integrate experimental data and uncover the possible regulatory mechanism of haematopoietic lineage commitments.

# 1. Introduction

Haematopoiesis is a dynamic process by which haematopoietic stem cells (HSCs) renew themselves to maintain the stem cell pool or generate multipotent progenitors (MPPs) that progressively differentiate to various specific lineages [1]. This extensively studied and well-characterized process has been orchestrated as a canonical haematopoietic lineage commitment roadmap of stepwise and hierarchical features. Although every individual progenitor can choose different cell fates, the total outputs of various cell types maintain balance. The dysfunction of lineage commitment can result in blood diseases, such as leukaemia and aplastic anaemia. Understanding the mechanisms that control the cell-fate commitments is of significant importance.

Despite the 'classical' hierarchical and bifurcated nature of the haematopoietic lineage commitments [1], conflicting observations have been made [2–4]. Distinct routes of erythro-megakaryocytic lineage generation have been observed [2]. In a single-cell study, progenitor cells were thought to have no definite direction of differentiation [3]. Moreover, the stem cell pool was also found to be phenotypically heterogeneous [5–7], which added another layer of complexity to the canonical roadmap model. It, therefore, remains challenging to reconstruct a precise differentiation roadmap. On the other hand, defects in transcription factors lead to abnormalities in differentiation [8]. One prominent example is that ectopic expression of specific genes may result in lineage conversions [9]. Understanding the regulatory mechanisms of reprogramming driven by ectopic gene expression might facilitate the exploration of novel therapeutic strategies.

A far-reaching metaphor of the epigenetic landscape was proposed by Waddington [10] to conceptualize cell differentiation as a ball rolling downhill. Stem cells roll down along the landscape shaped by the gene regulatory network and make cell-fate decisions at each saddle and finally reach different stable valleys that represent distinct cell types (electronic supplementary material, figure S1). The framework of the endogenous molecular–cellular network [11,12] provides a way to quantify the landscape, which has been demonstrated effective in several studies in both carcinogenesis [13–16] and development processes [17,18]. The main assumption is that molecular interactions and cell behaviours can be bridged by network dynamics, thus depicting the landscape of cellular states. Cellular phenotypes correspond to robust attractor states emerging from the dynamics of an endogenous network that collects interactions of key regulatory factors, and the changes of cellular phenotypes correlate to the state transitions on the landscape. The idea that attractor states of the gene regulatory network as cellular phenotypes could be traced back to Delbrück, Jacob and Monod, and was followed by Kauffman [19–21]. In the light of the endogenous network hypothesis, the process that stem cell differentiation into various cell types can be understood as the emergence of multiple robust states in the dynamics of an endogenous network and the transitions between these states. Recently, this assumption has also been validated by other studies of gene regulatory networks to interpret cell differentiation and reprogramming mechanisms using different quantification frameworks [22–29].

We attempt to construct an endogenous network and explore a landscape that orchestrates the robust states and their transition paths by quantifying network dynamics to understand haematopoietic lineage commitments. As shown in figure 1, in this study, we first constructed a primary endogenous network by collecting known gene–gene interaction knowledge from the literature. Then we refined the network to close the gap between simulation results and biological knowledge. A previously unrecognized interaction among the key transcription factors which is predicted essential by the dynamic model is supported by chromatin immunoprecipitation sequencing (ChIP-Seq) data and was integrated into the network to refine the model. A landscape of haematopoietic lineage commitments was thus discovered from the network dynamics. This landscape predicts previously under-represented cell states and their transition paths, and was further validated by re-analysing published single-cell RNA sequencing (scRNA-Seq) data. Importantly, the impacts of ectopic expression of genes can be simulated *in silico* by network perturbation analysis in this model, which would be helpful for studies on haematopoietic lineage conversions.

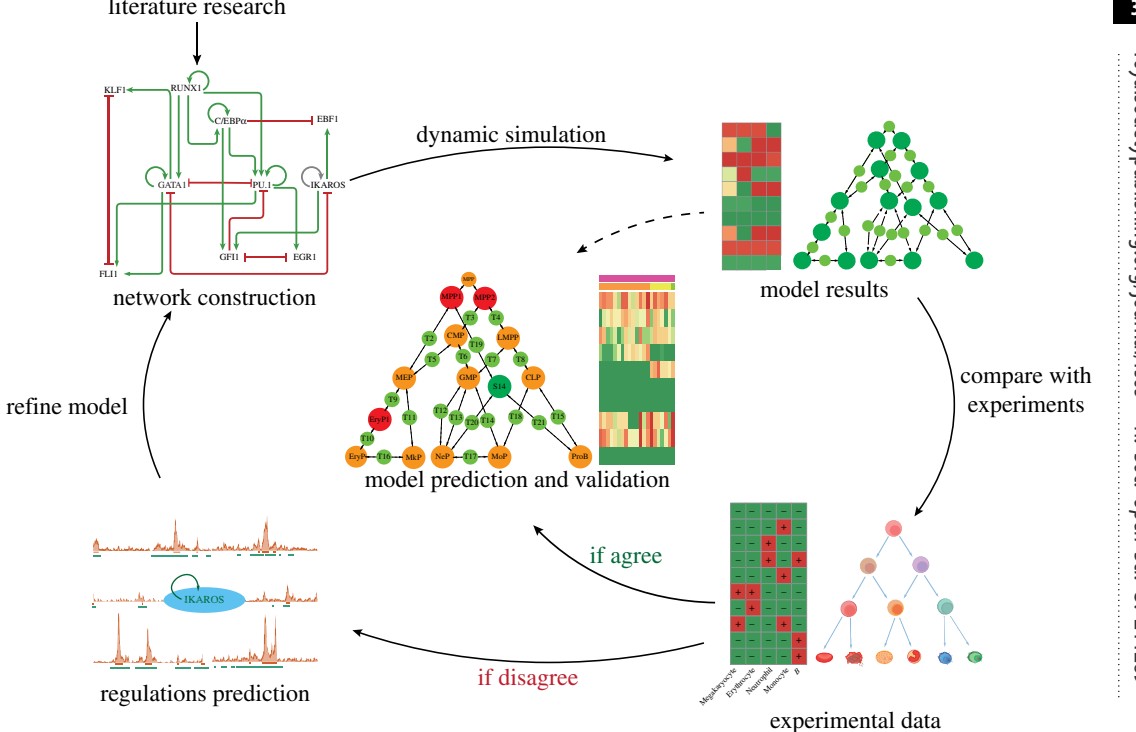

**Figure 1.** The workflow of endogenous network construction and the dynamic modelling. The endogenous network was constructed based on key transcription factors and is used to predict cell-fate decisions by its network dynamics. We first collected literature findings to identify core transcription factors related to haematopoietic lineage commitments and collected regulations among them. The attractor states and the state transition paths emerging from the network dynamics were compared with the known cell types and the canonical haematopoietic differentiation roadmap. If unmatched, the network was refined, and the new predicted regulation which is supported by high-throughput data is incorporated into the network. This process is iterated until the simulation results agree with the experimental data. The final model was used to predict the cellular states, lineage differentiation paths and other properties.

## 2. Results

### 2.1. Construction of the endogenous network

As shown in figure 2*a*, haematopoiesis is hierarchically organized by a series of progenitors generated by HSCs. HSCs lose self-renewal capacity and generate MPPs, which differentiate into progressively lineage-restricted progenitors. MPPs give rise to common myeloid progenitors (CMPs, which give rise to erythroid, megakaryocytic, granulocytic and macrophage progeny) and lymphoid-primed multipotential progenitors (LMPPs, which give rise to granulocytic, macrophage and lymphoid progeny). Haematopoietic lineage commitments involve complex regulations, among which the cross-antagonism between specific lineage-determining factors at each binary cell-fate branchpoint plays a pivotal role [30]. These genetic switch-like transcription factor pairs are the basis on which we constructed the network (figure 2*a*). The transcription factor RUNX1 is vital for the initiation of HSCs from the embryonic stage [31]. It also involves haematopoietic lineage commitment by its regulation of PU.1 [32], C/EBPα [33] and GATA1 [34]. The transcriptional cross-antagonism of GATA1 and PU.1 [35] instructs the alternative cell-fate choice between the Ery/Mk lineage and the myelo-lymphoid lineage [36]. IKAROS is crucial for the generation of the lymphoid lineage [37,38], and IKAROS/GATA1 functional antagonism lies at the branchpoint of the Ery/Mk lineage versus the lymphoid lineage [39]. The specification of lymphoid B cell versus granulocyte/monocyte fate occurs in the context of LMPPs. EBF1 specifies the B-cell lineage by interfering with myeloid regulators, with high-level expression of PU.1 and C/EBPα, to prevent alternative cell-fate choices. Genetic switch-like cross-antagonisms also dominate the secondary step of cell-fate determinants. In megakaryocyte–erythrocyte progenitors (MEPs), the erythrocyte versus megakaryocyte fate option is instructed by KLF1 and FLI1, a pair of counteracting repressors [40]. In granulocyte–monocyte progenitors (GMPs),

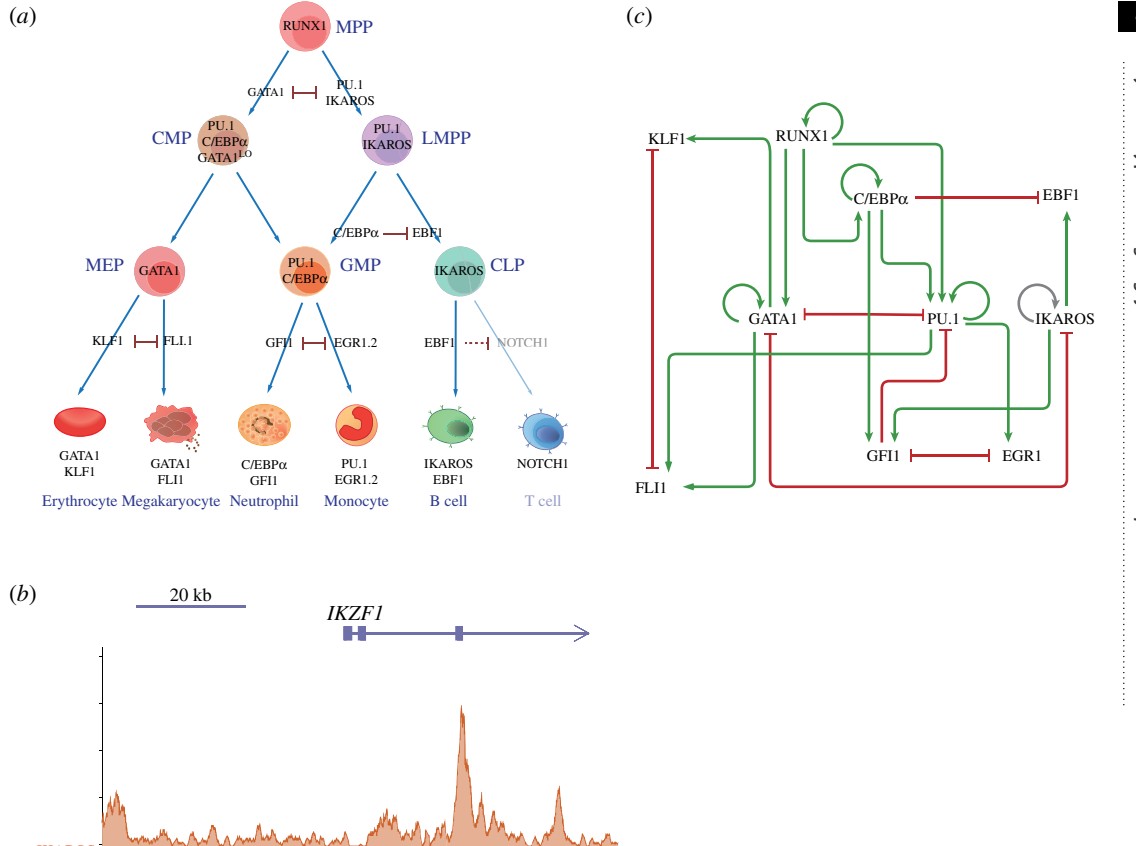

**Figure 2.** Construction of the endogenous network controlling haematopoietic lineage commitment cell-fate decisions. (*a*) The canonical haematopoietic lineage commitment roadmap and lineage-determining transcription factors work at the cell-fate branchpoint. The translucent part is not included in the network. (*b*) IKAROS ChIP-Seq signals and peaks (orange blocks under the signal) and H3K27AC peaks (green blocks under the signal) at the *IKZF1* gene loci. The arrow with boxes at the top presents the gene structure, with the arrow indicating the direction of transcription, and the boxes indicating exons. The gene location and structure information was obtained from the Ensemble dataset. (*c*) The core endogenous network controlling haematopoietic lineage commitments. The green arrows denote activations, the red blunt arrows denote inhibitions and the grey arrow denotes the predicted regulation.

increased PU.1 and C/EBPα modulate a regulatory circuit composed of counteracting repressors EGR1 and GFI1 [41], leading to the specification of a monocyte or granulocyte (neutrophil) lineage from the mixed lineage pattern; thus, EGR1 and GFI1 were chosen as components in our network. Lymphoid T-cell development will not be discussed since T cells migrate to the thymus with particular environmental signals. Auto-stimulations of genes stabilize cellular states [42], and in this network, RUNX1, PU.1, C/EBPα and GATA1 are experimentally self-activated [43–47]. In addition, other regulatory mechanisms coordinating these transcription factors were assembled in the network. The key transcription factors and their interactions are summarized in table 1.

This primary network was simulated to calculate attractor states (the computational procedure is shown in the following section and Methods section). The simulation results were compared with the experimental results (table 2), and we refined the network to close the gaps between them. In the refinement processes, we predicted one important positive auto-regulatory loop. One gap between simulation and experiments was that there is no state representing the expression pattern of monocytes (defined as PU.1 + EGR1 + FLI1+) and neutrophils (defined as C/EBPα + GFI1+) (electronic supplementary material, figure S2). Considering the important role of positive feedback for cellular state formation, we added auto-stimulation of the lymphoid transcription factor IKAROS into the network. Interestingly, the expected states emerged. We checked the ChIP-Seq data from the Encode Project [57] of the genome binding profiling of IKAROS in lymphoblastoid (figure 1*b*). *IKZF1* peaks were located both downstream of the first exon and upstream around 10 kb of the transcription start

**Table 1.** Regulatory relationships of the transcription factors in the core endogenous network collected from the literature.

| transcriptional factor | activator(s) | inhibitor(s) |
| --- | --- | --- |
| RUNX1 | RUNX1 [43] | |
| PU.1 | PU.1 [44], C/EBPα [48], RUNX 1[49] | GFI1 [41], GATA1 [50] |
| C/EBPα | C/EBP [45,46], RUNX1 [33] | |
| GFI1 | C/EBPα [51], IKAROS [41] | EGR1 [52] |
| EGR1 | PU.1 [52] | GFI1 [52] |
| GATA1 | GATA1 [47], RUNX1 [34] | PU.1 [53], IKAROS [39] |
| KLF1 | GATA1 [54] | FLI1 [40] |
| FLI1 | GATA1 [55], PU.1 [56] | KLF1 [40] |
| IKAROS | IKAROS [57] | GATA1 [39] |
| EBF1 | IKAROS [58] | C/EBPα [59] |

**Table 2.** The expression patterns of haematopoietic cells. '+' denotes the high-level expression of a gene, '−' denotes no expression or low expression and blank denotes that the gene expression pattern is not specific.

| | multiple potent progenitors | | | | | | specification lineages | | | | | |
| --- | --- | --- | --- | --- | --- | --- | --- | --- | --- | --- | --- | --- |
| | MPP | CMP | LMPP | MEP | GMP | CLP | ERY | MK | NE | MO | B | ref. |
| RUNX1 | + | + | | | | | | | | | | [60] |
| PU.1 | + | + | + | | + | + | | | | + | | [61,62] |
| C/EBPα | + | + | | − | + | − | − | − | + | | − | [61,63] |
| GFI1 | | | | | | | | | + | | + | [41,64] |
| EGR1 | | | | | | | | | | + | | [52] |
| GATA1 | | | − | + | − | − | + | + | − | − | − | [61] |
| KLF1 | − | − | − | | | | + | | | | − | [65] |
| FLI1 | + | | | | | | − | + | | + | | [66,67] |
| IKAROS | + | | + | − | − | + | − | − | − | − | + | [68,69] |
| EBF1 | − | − | − | − | − | | − | − | − | − | + | [70] |

site, which also had H3K27AC modifications, implying that positive autoregulation of IKAROS is a reasonable prediction. This regulation is presented in figure 2c as a grey arrow.

The details of the whole network refinement process were shown in the electronic supplementary material. The refined network is shown in figure 2c. The modelled prediction of this network was consistent with the experimental data. This refined model was used for the following dynamic simulation.

## 2.2. From molecular network to cellular phenotype

### 2.2.1. Robust states and their transition paths

Multiple feedbacks coordinate the regulatory network, which add 'constraints' to the expression status of the transcription factors. Due to these constraints, only a limited number of gene expression patterns are steady, which do not exert driving force to change present states [71]. Mathematically, these states are fixed points of differential equations of a dynamical system [72]. A fixed point is an attractor state if it is stable under tiny perturbations, while unstable points are called saddle points. Because of this stability, attractors are considered to correspond to stable cell types in a biological system. Saddle points connect to two or more specific states (figure 3a), and they were named transition states here. These states intermediate spontaneous transitions among attractor states driven by noise. They can be interpreted as unstable cell states biologically.

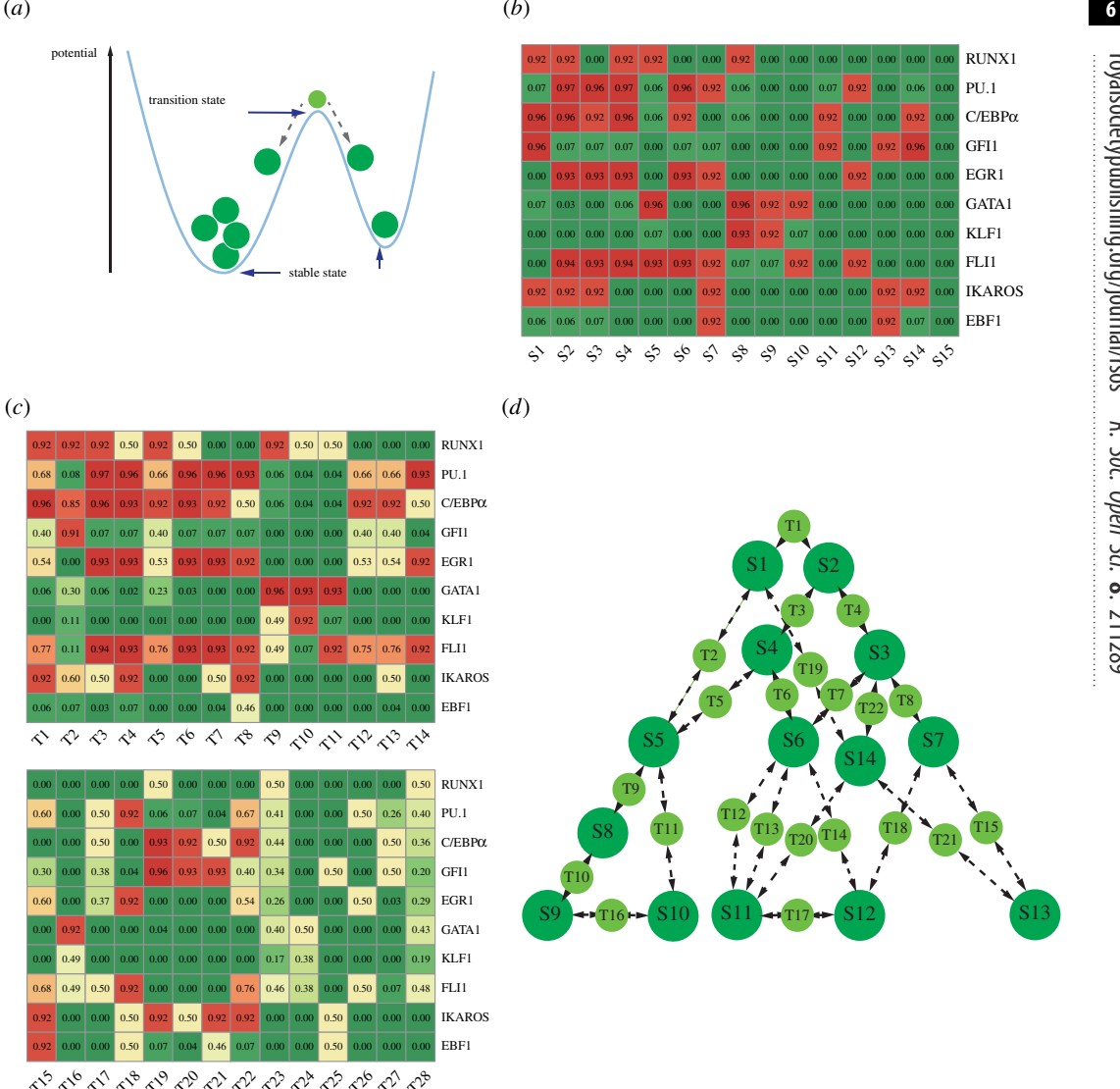

**Figure 3.** Attractor states, transition states and their transition paths. (*a*) Attractor states correspond to valleys in the landscape where the potential energy is low. Transition states correspond to saddles in the landscape where the potential energy is relatively higher, and small perturbations can lead to the point flow to attractor states. (*b*) Attractor states obtained from the network dynamics. Each column is a calculated state. The expression levels of the transcription factors in each state are shown. '1' represents the maximal expression; '0' denotes that its level is not sufficient to affect its target genes significantly. (*c*) Transition states obtained from the network dynamics. (*d*) The topological connection graph representing attractor states, transition states and their transition paths. The large forest-green circles denote attractor states, while the small light-green circles denote transition states. The arrows denote transitions between states generated by small perturbations. S15 without any transcription factor expression is not shown in the landscape.

Fifteen attractor states (figure 3*b*) and twenty-eight transition states (figure 3*c*) were obtained by simulations of trajectories of ordinary differential equations (ODEs) for the network dynamics from random initial conditions. By perturbing each transition state (electronic supplementary material, figure S5), the transition paths among states are obtained. The attractor states, transition states and their transition paths are mapped to a topological connection graph, which reflects the multi-stability and state transitions of the system (figure 2*d*). Additionally, 12 saddle points connecting more than two stable states were obtained in the system and were named hyper-transition states (electronic supplementary material, figure S4). They will not be discussed in detail in this study.

Furthermore, we evaluated the robustness of the dynamical model by simulating ODEs under different parameter conditions (electronic supplementary material, table S1). All fixed points were conserved, with only subtle variations in the values. Moreover, Boolean logical rules were used to

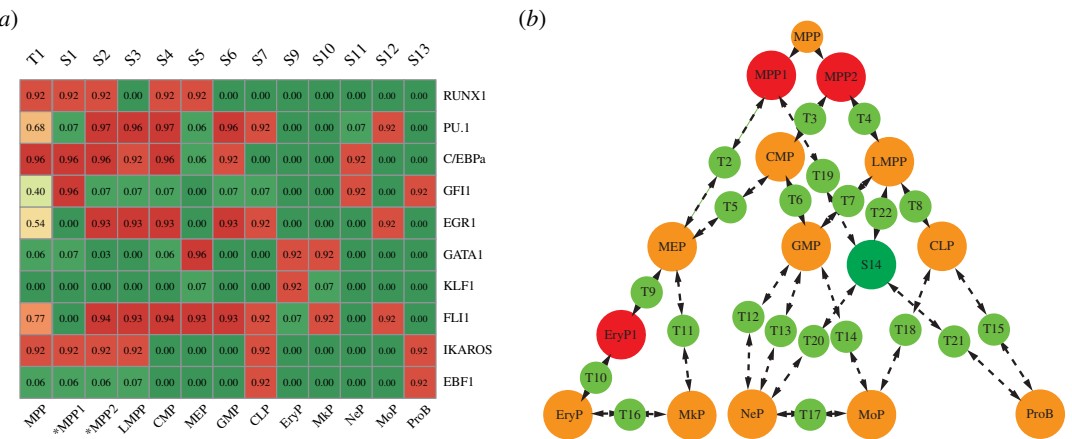

**Figure 4.** Cell types and the differentiation landscape predicted by the model. (*a*) The robust states of the well-defined cell types and predicted cell states are calculated by the network dynamics. * Denotes the predicted cell states. (*b*) The canonical differentiation roadmap is well reproduced by the model-predicted landscape. In addition, previously unrecognized cell states are predicted by the model. The orange circles represent the well-characterized cell types, the red circles represent the predicted cell stages and the green circles represent other unclassified states.

describe the gene interactions in the network. The Boolean model is a parameter-free method to obtain attractor states of a dynamical system and has been widely used in gene regulatory network studies [29,73–75]. We compared the attractor states obtained by the Boolean model with those obtained by the ODEs, and found a one-to-one correspondence between them, in the sense of the levels (high/low) of each gene of every attractor state (figure 3*b*; electronic supplementary material, figure S8). The consistency of these results suggests that the attractor states are intrinsically robust, and exist under a large range of parameters and different forms of the model.

## 2.2.2. The well-known cell types in haematopoietic lineage commitments are reproduced, and model-predicted novel cell types are revealed from the landscape

We next examined whether these obtained states reflect known haematopoietic cell types. A state identified as a well-characterized cell type should have a compatible expression pattern and consistent differentiation potential directions with known biological knowledge (table 2). The states representing unipotent progenitors such as erythrocytes, megakaryocytes, neutrophils, monocytes, and B cells were obtained in the network as stable states S9–S13 (figure 4*a*). The MEP state was captured as S5 [34]. Analogically, the GMP differentiates into granulocytes (neutrophils) or monocytes, which express the GMP-specific genes PU.1 and C/EBPα. The GMP state was captured by S6. S7 immediately differentiates into the Pro-B-cell state. In addition to IKAROS and EBF1, it also expresses relatively high levels of PU.1 and was identified as a common lymphoid progenitor (CLP) state. LMPP [76] and CMP [61] can give rise to all lymphoid lineages or myeloid lineages. According to their differentiation potential directions, S3 and S4 were identified as the LMPP state and CMP state, respectively. The expression pattern of state T1 was the same as that of MPPs, which was, therefore, identified as an MPP state. In summary, all the well-known cell types are reproduced.

Moreover, two previously uncharacterized progenitors are predicted from the landscape. Interestingly, MPP state T1 directly connected S1 and S2, indicating that they retain similar characteristics. When T1 transits to S1, the level of PU.1 descends, and then S1 immediately transits to MEP, which is accompanied by the increase of GATA1 expression. We predicted S1 as an Ery/Mk-biased multipotent progenitor state (MPP1). For S2, the PU.1 level increased in this state, implying its preference towards Ne/Mo/lymphoid lineages but not the Ery/Mk lineage, although it remained multipotent. Consequently, it is predicted to be an LMPP-biased multipotent progenitor named as MPP2. S8 is the state in which an MEP is intermediated to an erythrocyte and is labelled as EryP1.

Finally, the inferred cell states were mapped to the state connection graph and we thus obtained the haematopoietic lineage differentiation landscape (figure 4*b*). The landscape presents the differentiation process shown in the canonical roadmap. In addition, the model predicted that MPP1 and MPP2 intermediate differentiation paths, which indicates possible mechanisms of haematopoietic lineage commitments.

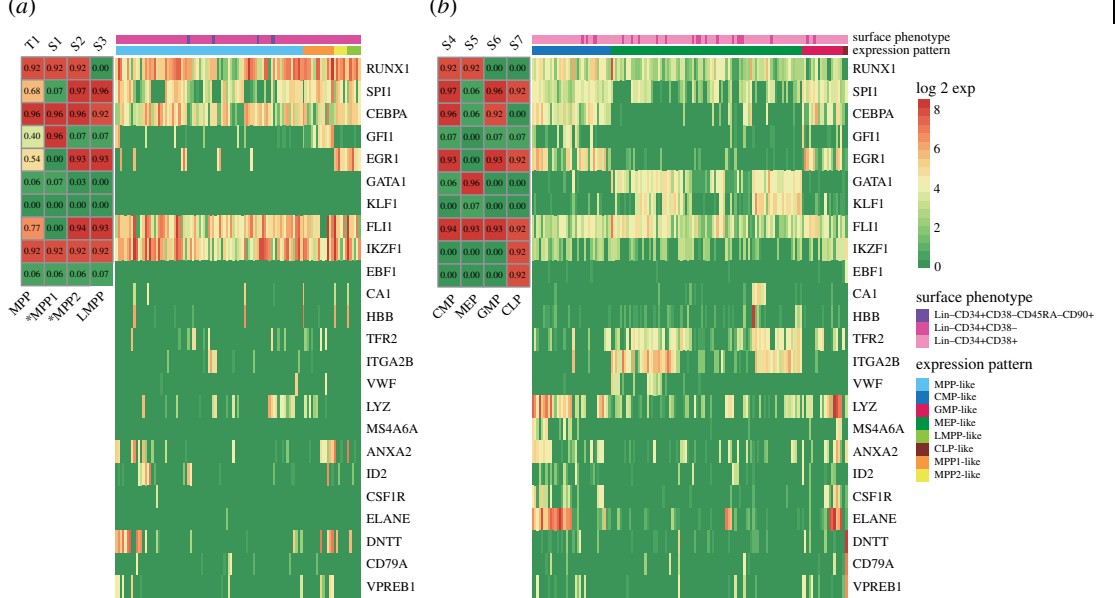

**Figure 5.** Cell types predicted in the dynamical model reproduced from scRNA-Seq data. (*a*) Verification of the predicted CD38- progenitor MPP-like, MPP1-like, MPP2-like and LMPP-like states in the scRNA-Seq data. (*b*) Verification of the predicted more differentiated progenitors that are marked by CD38+, including the CMP-like, MEP-like, GMP-like and CLP-like states in the scRNA-Seq data. The profiles of the core transcription factors of the network as well as 14 lineage-specific genes that imply cell types are shown in the heatmap.

## 2.3. Validation of the modelling results

We used the scRNA-Seq data of human bone marrow cells [3] to validate the model-predicted MPP1 and MPP2 states and known cell types MPP, LMPP, CMP, MEP, GMP and CLP states. It has been reported that HSCs and their immediate progenies reside in the Lin-CD34+CD38- compartment of bone marrow cells, whereas more differentiated progenitors reside in the Lin-CD34+CD38+ compartment [4]. We first analysed the scRNA-Seq data of Lin-CD34+CD38- cells, examining the expression profile of the 10 core transcription factors as well as 14 lineage-specific genes (figure 5*a*). Specifically, we characterized whether there were cells with a specific expression pattern corresponding to the stable state we obtained. Cells with the expression pattern we defined as MPP partially resided in the Lin-CD34+ CD38-CD45R-CD90+ compartment, which enriched the population of HSCs. As shown in figure 4*a*, cells with MPP and HSC markers were mapped to our model-defined MPP state, suggesting that HSCs and MPPs were indistinguishable based on our model. In fact, these two types share a broadly similar expression profile [77]. Not only MPP state T1 and LMPP state S3, but also the predicted S1 and S2 are reproduced from the Lin-CD34+CD38- bone marrow cells. The 14 lineage-specific genes were mostly expressed at a very low level, which agrees with their low differentiated characteristics.

We then identified the S4–S7 cell states from the Lin-CD34+ bone marrow scRNA-Seq data, most of which resided in the CD38+ compartment (figure 5*b*), consistent with previous experimental findings [78]. In addition, other lineage-specific genes also indicated the lineage potential of progenitors. In the MEP-like state, we confirmed that erythrocyte-specific genes CA1, HBB, and TFR2 and megakaryocyte-specific genes IRGA2B and VWF were highly expressed. In the GMP-like state, the specific genes ID2 and CSF1R were expressed, while LYZ, MS4A6A and ANXA2 specifically in monocytes and ELANE in neutrophils were also expressed. Similarly, the B-cell lineage genes DNTT, CD79A and VPREB1 were highly expressed in the CLP-like state in our model.

## 2.4. Dynamical analysis of distinct routes of Ery/Mk lineage development

The unequal output of the amount of the granulocyte/macrophage lineage and the megakaryocyte/ erythrocyte lineage generated by the CMP population [2,79] suggests that the Ery/Mk lineage may separate from myelo-lymphoid lineages earlier than CMP generation [2,80,81]. We then deciphered the developmental routes of the Ery/Mk lineage from our model. On the landscape, distinct routes bridge the MPP state and the MEP state (figure 6*a*). Thus, two independent Ery/Mk progenitor generation

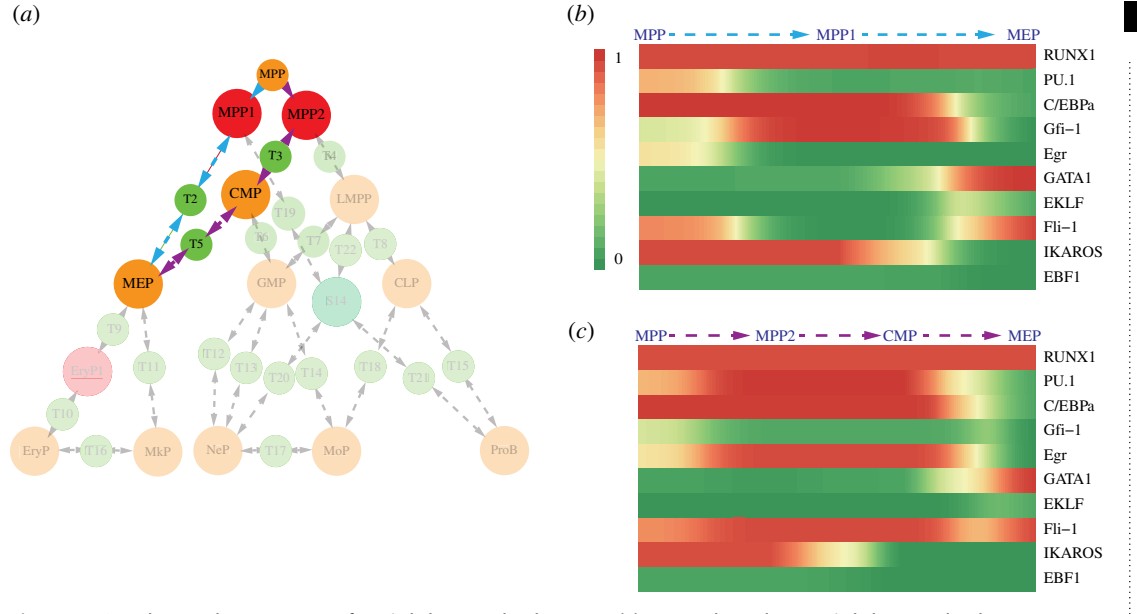

**Figure 6.** Deciphering distinct routes of Ery/Mk lineage development. (*a*) Two independent Ery/Mk lineage development routes emerge in the landscape. One route is intermediated by the MPP1 state (blue arrows), while the other route is intermediated by the MPP2 and CMP states (purple arrows). (*b*,*c*) Transcription factor fluctuation during lineage generation predicted by the LAP: (*b*) in the MPP1 intermediated route; (*c*) in the MPP2 and CMP intermediated route.

routes were predicted: one route is intermediated by MPP1 states, and another route is intermediated by MPP2 and CMP states.

To determine the potential regulatory mechanism of lineage commitment, we predicted the dynamical fluctuations of the expression levels of transcription factors during Ery/Mk lineage generation. Transitions among attractor states are triggered by noise. In a biological system, the noise can be transcriptional noise or fluctuating signals [82]. Transitions are probabilistic in the stochastically perturbed dynamical system. The least action path (LAP) method [83] provides an algorithm to obtain the most likely transition by minimizing the action along the transition path. The action appears as an exponentially decaying factor associated with the probability of a transition.

We predicted transcription factor fluctuations during haematopoietic lineage differentiation by the LAP method. In the first route mediated by MPP1 (figure 6*b*), PU.1 promptly decreased when the differentiation of MPP started, followed by an increase in GATA1 expression, suggesting that in this route, the antagonism of GATA1 and PU.1 plays a pivotal role in orienting MPPs differentiated into Ery/Mk lineages or other myeloid/lymphoid lineages. In the second route mediated by MPP2 and CMP (figure 6*c*), IKAROS decreased at first; subsequently, PU.1 decreased, while GATA1 increased. This result implies that in this route, LMPPs that separate myeloid lineages are predominantly instructed by IKAROS. This separation is the first step of MPP differentiation. Thereafter, the antagonism of GATA1 and PU.1 directed the second branching of the CMP context.

## 2.5. Network perturbations predict the effects of ectopic gene expression

Next, we simulated the effects of ectopic gene expression during haematopoietic lineage commitments by perturbations of the network. For the knockout experiments of key regulators, we simulated the loss function of the gene in the model and then calculated the attractor states and transition states of the network dynamics. The effects of various gene knockout experiments are analysed in table 3. Our model predicted that the loss of Runx1 resulted in no definite lineage emergence. Specifically, neither Ery/Mk, lymphoid MPPs nor any Mo/Neu lineage states could be produced based on our model prediction, which was in consistence with the reported effect of haematopoiesis defects by *in vivo* knockout of RUNX1 [84]. It was reported that *Spi1* (gene encodes PU.1) knockout results in the loss of myeloid and lymphoid lineage MPPs [85]. Consistent with these experimental findings, only the MEP and Ery/Mk states were obtained in our model when simulated PU.1 knockout. *Cebpa* knockout resulted in the blockage of GMP and Mo/Neu lineages [63]. In the simulation of C/EBPα knockout, there was no state of GMP or Mo/Neu lineages, which agreed with experimental results. Likewise,

**Table 3.** Summary of the impact of the selected gene knockout in the experiments and in the model.

| experiment model | gene ectopic expression | model (states emerging) | knockout experiment | references |
|---|---|---|---|---|
| KO mice | Runx1 KO | no definite lineage | no haematopoiesis | [84] |
| MPP | Spi1 KO | Ery/Mk lineages | no mye/lym development | [85] |
| KO mice | Cebpα KO | Ery/Mk/B lineages | block GMP | [63] |
| KO mice | Gata1 KO | Mo/Ne/B lineages | blocked E/Mk | [86,87] |
| KO mice | Ikzf1 KO | Ery/Mk/Mo/Ne lineages | no lymphoid cells | [68,88] |
| KO mice | Ebf1 KO | no B-cell lineages | no B-cell differentiation | [89] |

*Gata1*, *Ikzf1* (gene encodes IKAROS) and *Ebf1* knockout lead to the loss of their corresponding lineages, i.e. the Ery/Mk, lymphoid and B-cell lineages, respectively [68,86–89]. These experiments were simulated in the model. The results of the simulations and experiments are consistent.

We next simulated the conversions between distinct lineages induced by specific gene over-expression. Setting the value of GATA1 to the maximum in the CLP, Pro-B, CMP and GMP states, all of these states converted to the GATA1 + FLI1+ or GATA1 + FLI1 + KLF1+ states, showing an Ery/Mk-like pattern (figure 7*a–d* and table 4). This outcome agrees with documented experimental results [90,91]. Likewise, the conversion into the granulocyte/macrophage lineage of CLPs [92], MEPs [92] and Pro-B cells [59,92] can be induced by the over-expression of C/EBPα. They were simulated in the model by setting the value of C/EBPα to maximum (figure 7*e–g* and table 4). The induction of C/EBPα in the MEP causes the state to convert to C/EBPα + GFI1+, an NeP-like state. In CLP and Pro-B, the state converts to an LMPP-like expression pattern and a C/EBPα + GFI1 + IKAROS+ expression pattern, respectively. Various lineage-determining transcription factors are present in these states, presenting MPP-like patterns, which may suggest a dedifferentiation event during these lineage conversions.

# 3. Discussion

Despite the extensive use of molecular and cellular technologies and the high-throughput assays in cell lineage commitment studies, understanding the essential regulatory mechanisms and principles of cell-fate decisions and using them to guide the experimental design is still challenging. In the present work, we use the endogenous network modelling to address the regulatory mechanism of the haematopoietic lineage commitments. In addition to the reproduction of known cellular behaviours in this process, the model can also make new predictions.

We constructed the endogenous network, which is mainly based on reported causal regulations that govern haematopoietic lineage commitment. In two aspects, it is called a coarse-grained model: (i) a minimal-scale network is constructed. To construct an endogenous network of the haematopoietic lineage commitments controlling, we focused on genetic switch-like gene pairs, which work antagonistically at each cell-fate branchpoint to instruct lineage specifications. This idea was obtained from the previous study on the Phage λ genetic switch [93]. The endogenous network composed of CI and Cro and their cross-antagonism explained experimental data quantitatively regarding the choice of alternative survival modes of Phage λ, lytic or lysogenic. (ii) A coarse-grained description of interactions between regulatory factors is used, which are considered simply as 'activation' or 'inhibition', both in the graph of the network and in the mathematical model. Considering the lack of details regarding regulatory mechanisms and biochemical parameters, this coarse-grained implementation is indispensable. In fact, by using different coefficients of the Hill function, we found that our model is parameter-independent. Attractor states emerging from the network are quite robust in parameter variations and are mostly determined by the architecture of the network. Our results show that although with incomplete knowledge, cell lineage commitments can be explained by the landscape-orchestrated robust states and transitions among them. Specifically, the gene expression patterns of these robust states and the multi-step hierarchical branching transitions obtained from simulation results agree with the experimental data.

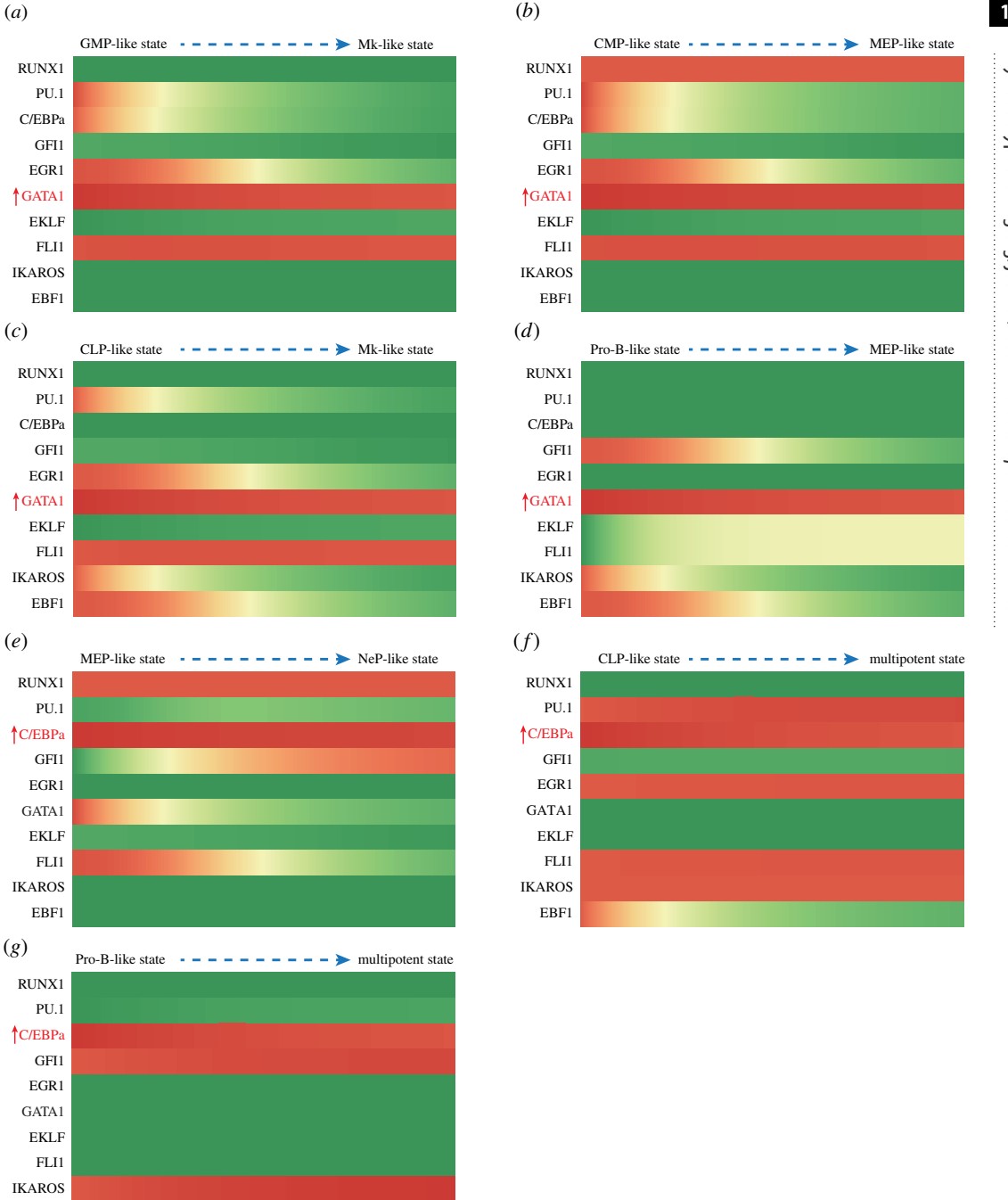

**Figure 7.** Gene expression fluctuations during lineage conversion upon specific gene induction. (*a–d*) Induction of GATA1 (*a*) in the GMP-like state, (*b*) in the CMP-like state, (*c*) in the CLP-like state and (*d*) in the Pro-B-like state. (*e–g*) Induction of C/EBPα (*e*) in the MEP-like state, (*f*) in the CLP-like state and (*g*) in the Pro-B-like state.

The present model leads to a certain number of predictions. First, the model predicted the auto-stimulation of IKAROS, which is essential for reproducing known cell types in modelling, and the rationality of this regulation was proven by an investigation of the ChIP-Seq data. Second, the landscape orchestrated robust states and their transitions. Both well-characterized cell types and differentiation routes that had not yet been recognized in the canonical hierarchy are shown. The heterogeneity of the stem cell pool was exactly embodied in the model. The MPP, MPP1 and MPP2 states had distinct lineage biases that were caused by the different transcription factors 'winning out' in the switch-like genetic antagonisms. These results were further validated by scRNA-Seq data. Third, the landscape showed a distinct source of the Ery/Mk lineage. The fluctuations of the

**Table 4.** Summary of the impact of the selected gene induction in specific progenitors in the experiments and in the model.

| cell type | gene ectopic expression | model (state convert to) | experiment | references |
|---|---|---|---|---|
| CMP | GATA1 activation | GATA1+FLI1+ | Ery/Mk | [90,91] |
| CLP | GATA1 activation | GATA1+FLI1+ | Ery/Mk | [90] |
| GMP | GATA1 activation | RUNX1+GATA1+FLI1+ | Ery/Mk | [90] |
| Pro-B | GATA1 activation | GATA1+FLI1+KLF1+ | Ery/Mk | [90] |
| MEP | C/EBPα activation | RUNX1+C/EBPα+GFI1+ | Mo/Ne | [92] |
| CLP | C/EBPα activation | PU.1+C/EBPα+EGR1+FLI1+IKAROS+ | Mo/Ne | [92] |
| Pro-B | C/EBPα activation | C/EBPα+GFI1+IKAROS+ | Mo/Ne | [59,92] |

transcription factors during these two routes of state transitions were simulated. The dynamics of these processes suggested different switch-like gene antagonisms; separately, GATA1/PU.1 or GATA1/IKAROS played a pivotal role in the origin of the MPP differentiation. Furthermore, we performed documented ectopic gene expression experiments *in silico*, including knockouts of Runx1, Spi1, Cebpa, Gata1, Ikzf1 and Ebf1, as well as over-expressions of C/EBPα and GATA1 in different cell types. The results of the model were in good agreement with those of the experiments. They may imply events such as dedifferentiation during lineage conversions, but further proof is needed in future work. This modelling framework may provide an effective and efficient method for cell reprogramming studies.

Several dynamical models of haematopoietic lineage specification have been reported [29,94]. Compared with those works, our model predicts a more complete haematopoietic lineage differentiation roadmap. Using continuous variables and the ODE model, our model captures more precise information for the dynamical system, which shows its advantages compared with the discrete Boolean model. In the work of Collombet *et al.* [29], important molecular pathways were also assembled into the regulatory network. In the future, more transcription factors, cytokines and signalling pathways involved in this process can be incorporated into the model to simulate the exogenous signalling and to improve the quality of predictions. On the other hand, from the perspective of the endogenous network, the network is hierarchical and could be composed of relatively independent functional modules such as the cell cycle, metabolism and apoptosis. Our group [18,95] has provided an example in the study of myelopoiesis.

Our work shows that even without detailed information on molecular interactions, we can construct a dynamical model to predict cell behaviours in haematopoietic lineage commitments. This multistable dynamical system has properties of predictability and extensibility. More regulatory interactions and detailed properties can be incorporated into the network to improve the quality of modelling predictions or extend them to other haemetopoietic cell types. Furthermore, this modelling framework can be used as a template to study cell lineage commitments and conversions.

# 4. Methods

## 4.1. ChIP-Seq data analysis

The bigwig file was converted into a bedGraph file using bigWigToBedGraph from Conda, and tracks were extracted by bedtools. Signals in the 10 kb regions upstream and downstream of the start site of the first exon were extracted for each gene. Gene information was obtained from www.ensembl.org. Peaks to gene domain associations and gene structure were presented by the R package ggplot2.

## 4.2. Network modelling

The dynamical model for the haematopoietic lineage commitment network was constructed based on the proposed framework of the endogenous network [12]. Since details of the interactions between the factors are not available in most cases, a coarse-grained modelling approach was employed. The interactions between the factors of the network were described simply as activation or inhibition. A set of ODEs was used to describe the interactions in the network, by which the attractor states, transition states and their transition paths were calculated. Boolean logical rules were also used to obtain attractor states to demonstrate that the model is parameter-independent.

### 4.2.1. Ordinary differential equations

Assuming that transcription factors and RNA polymerase binding/unbinding are synchronous with the other reactions engaged, such as translation and decay, formalisms from steady-state models can be employed. The production rate of factors can be described by a chemical rate equation that consists of the production rate and degradation rate [96–99]

$$\frac{\mathrm{d}x_i}{\mathrm{d}t} = f(x_i) - \frac{x_i}{\tau_x},$$

(4.1)

where $x_i$ is the concentration/activity of the factor, $f(x_i)$ is the production rate of $x_i$ under the regulations of other factors, and $x_i / \tau_x$ is a linear degradation of $x_i$. The degradation constant is set to be normalized for all factors, i.e. $\tau_x = 1$.

The molecular interactions among factors were modelled by the Hill function, which has been used for cooperative interactions [99–102]

$$f_{\mathrm{activation}} = \frac{a[\mathrm{activator}]^n}{1 + a[\mathrm{activator}]^n}$$

(4.2)

and

$$f_{\mathrm{inhibition}} = \frac{1}{1 + a[\mathrm{inhibitor}]^n}.$$

(4.3)

The parameter $n$ indicates the Hill coefficient of the Hill function, while $a$ indicates the apparent dissociation constant. The value of each factor $x_i$ was normalized from 0 to 1. We can reasonably assume that when $[\mathrm{activator}] = 0$, then $f_{\mathrm{activation}} = 0$; when $[\mathrm{activator}] = 1$; then $f_{\mathrm{activation}} = 1$; when $[\mathrm{activator}] = 1/2$, then $f_{\mathrm{activation}} = 1/2$. Thus, we can deduce that $a = 2^n$.

The following consist of two terms

$$f(x_i) = \frac{a \cdot \sum_{j \in \mathrm{activators}} x_j^n}{1 + a \cdot \sum_{j \in \mathrm{activators}} x_j^n} \cdot \frac{1}{1 + a \cdot \sum_{k \in \mathrm{inhibitors}} x_k^n}.$$

(4.4)

To sum up, the rate of change for each factor is

$$\frac{\mathrm{d}x_i}{\mathrm{d}t} = \frac{a \cdot \sum_{j \in \mathrm{activators}} x_j^n}{1 + a \cdot \sum_{j \in \mathrm{activators}} x_j^n} \cdot \frac{1}{1 + a \cdot \sum_{k \in \mathrm{inhibitors}} x_k^n}.$$

(4.5)

All 10 equations are listed in the electronic supplementary material.

We turned $n$ and $a$ ($n = 3, 4, \dots, 10, a = 8, 16, \dots, 1024$) (see electronic supplementary material, table S1) in the equations to test the consistency of the attractor states, to evaluate the sensitivity of the model to parameters. The numerical results shown in the Results section are the calculation under the parameter $n = 4$ and $a = 16$.

#### 4.2.1.1. Attractor states and transition states

The fixed points are the points $x$ that satisfy $\mathrm{d}x/\mathrm{d}t = 0$. These points were calculated by the MATLAB function 'fsolve', which is a numerical method for solving nonlinear differential equations based on the Newton iteration algorithm. When calculating fixed points, an initial point is needed to execute the iteration algorithm. A large number of random initial points were used to compute fixed points. The random initial point (a vector whose dimension is the number of factors in the network) is uniformly distributed in the interval [0, 1], which is generated by the MATLAB function 'rand'.

The fixed points of the system can be categorized by the signs of the eigenvalues of their Jacobian matrix. If the real parts of the eigenvalues are all negative, the fixed point is identified as an attractor state. It is a local stable state. If one or more eigenvalues have a positive real part, the fixed point is an unstable state, a.k.a. a saddle point. It will transit to different states under perturbation. With respect to the number of eigenvalues with a positive real part, we defined an unstable state with a single eigenvalue of the positive real part as a transition state, and those with two or more eigenvalues of the positive real part as hyper-transition states. When perturbed, the transition state flows to the other two states, while the hyper-transition states flow to more than two states. The hyper-transition states were not discussed in this work.

A series of random initial points crossing several orders of magnitude are executed in the computation. If no new attractors can be found, the attractors with a relatively large attractive basin,

which were considered as biologically stable states, have been obtained. Here, we used $10^5$ random initial points for the numerical computations, and no new attractor states were found when performing $10^8$ random initial points. This result indicated that $10^5$ computations are sufficient to obtain major attractor states.

### 4.2.1.2. Transition paths among states

An algorithm reported in [17] is designed to find the transitions among the stable states and transition states. The random perturbation vectors were generated by linear combinations of the eigenvectors of the Jacobian matrix corresponding to the eigenvalues that have positive real parts. That is, perturbations were performed on the unstable subspace of the transition states. The amplitude of a perturbation $\Delta p$ is restricted by the inner product $\langle \Delta p \cdot \Delta p \rangle < \delta, \delta = 1 \times 10^{-6}$. Exerting perturbation on each transition state, the point will flow to different states along the trajectory $\{x_i\}$. A state $x$ is defined as a terminal state if there exists $x_i$, satisfying $\langle x_i - x, \ x_i - x \rangle < \varepsilon, \ \varepsilon = 1 \times 10^{-8}$. A total of 1000 perturbations were performed for each transition state. The detailed results are shown in the electronic supplementary material, figure S5.

### 4.2.2. Boolean logical rules

The Boolean dynamical method is a discrete and parameter-free method that can be used to acquire the architectural properties of the network, and it has been widely used in modelling biological processes [29,73–75]. This simplified approach was applied as a reference to the ODE model.

In the Boolean model, activations and inhibitions are combined into logical rules by the Boolean operators AND, OR and NOT. The state of the network updates iteratively under the constraints of the Boolean logical rules. The full list of Boolean rules of the network can be viewed in the electronic supplementary material.

The operation of the Boolean rules was converted into matrix operations, and the matrix can be viewed in the electronic supplementary material. Each node $t$ has only two states, $S(t) = 1$ and $S(t) = 0$, representing the high expression level state and the low expression level state. Given the state value $S(t)$ at the time $t$, the state $S(t + 1)$ is given by $\mathbf{S}(t + 1) = \mathbf{R}S(t)$, where $\mathbf{R}$ is the regulation matrix. The values of the factors evolving over time are given by

$$S_i(t + 1) = \begin{cases} 1 & \sum_j r_{ij} S_j(t) > 0 \\ 0 & \text{otherwise} \end{cases}, \tag{4.6}$$

where

$$r_{ij} = \begin{cases} 0 & x_j \notin \text{activators/inhibitors of } x_i \\ 1 & x_j \in \text{activators of } x_i \\ -100 & x_j \in \text{inhibitors of } x_i \end{cases}. \tag{4.7}$$

The stable states should satisfy $S(t + 1) = S(t)$.

Here, all $2^{10} = 1024$ states of the 10-node network were traversed as the initial point (electronic supplementary material, figure S6). After iterations of Boolean rules, the attractor states of the network dynamics were found (electronic supplementary material, figure S7). We obtained 15 attractors, which is consistent with the stable state of the ODE model (electronic supplementary material, figure S8). In fact, when the Hill coefficient of ODEs $n \to +\infty$, the Hill equation corresponds to the Boolean rules (electronic supplementary material, figure S9). Details about the model and its connection with the ODE model are discussed in the electronic supplementary material.

## 4.3. Modelling results validation

The single-cell RNA-Seq data of Lin-CD34+ human bone marrow cells (GEO: GSE75478) [3] were used to validate the modelling results (figure 5). The RNA counts and surface marker data of single cells were used. The expression data are the value of $\text{Log}_2(\text{counts} + 1)$ for each gene. The expression data for 10 transcription factors of the network were used to classify cells into various model-predicted states of cell type. The complete table of the expression data of 10 transcription factors and 14 lineage-specific genes, surface phenotypes and predicted states of single cells can be viewed in the electronic supplementary material, file S2. Heatmaps were generated using the R software, package pheatmap.

## 4.4. Least action path analysis

The LAP method [83] calculates the most likely transition path by minimizing the energy cost, that is, the least action, along with the state transition under the stochastic dynamical system. The LAP is the $x$ that minimizes the action function $S(x)$, which is given by

$$S_{T_1 T_2}(x) = \frac{1}{4} \int_{T_1}^{T_2} \langle \dot{x} - f(x), D^{-1}[\dot{x} - f(x)] \rangle \, ds. \tag{4.8}$$

This was discretized as

$$S_{T_1 T_2}(x(t)) = \frac{1}{4} \Delta t \sum_{k=1}^{N} \sum_{i=1}^{M} \left\| \frac{x_i^{k+1} - x_i^k}{\Delta t} - \frac{f_i^{k+1} + f_i^k}{2} \right\|^2. \tag{4.9}$$

Given the time interval $[T_1, T_2]$ of the trajectory, it was divided into $N$ equal subintervals, and $T_1 = t_1 < t_2 < \ldots < t_{N+1} = T_2$.

The minima of the action functional are calculated by the MATLAB function fminunc. The line segments connecting the initial states and terminal states are used as initial paths. $T = 10$ and $N = 200$ were used. For larger $T$ and $N$, the result was convergent.

## 4.5. Genetic perturbations simulation

For the gene knockout simulations, we configured the value of the knockout gene as '0' and removed all of its interactions in the network. The corresponding ODEs of this network were established and the attractor states were calculated by $10^5$ random initial points. The obtained attractor states for each simulation can be viewed in electronic supplementary material, table S2.

For the gene ectopic over-expression simulations, we set the value of the over-expression gene as the maximum value '1' in the attractor point corresponding to the cell type as the initial point. Then, we calculated the trajectory to simulate lineage conversion by Euler's method: first, a random vector $x_0$ was set; then, $x_0$ was iterated following $x_{i+1} = x_i + \Delta t \times f(x_i)$ and the trajectory is $\{x_i\}$. A state $x_i$ is convergent if satisfying $|x_{i+1} - x_i| < \varepsilon$, $|f(x_i)| < \delta$ after a considerable number of iterations. Finally, $x_i$ was recorded as the result of lineage conversion.

We set the step size $\Delta t$ of the iteration as 0.01, $\varepsilon = 1 \times 10^{-8}$, and $\delta = 1 \times 10^{-12}$. The results of every simulation were converged, and the values of the first 3000 iterations for each simulation are shown in a heatmap as the cell transition trajectories.

Data accessibility. The accession numbers of the ChIP-Seq data used in the manuscript are GSM935442 and GSM2040756. The accession number of the scRNA-Seq data is GSE75478. The processed data and the input file for simulation have been provided in the electronic supplementary material [103].

Authors' contributions. M.W. carried out modelling work, participated in data analysis and the design of the study, and drafted the manuscript; J.W. participated in the design of the study, participated in the design of the model and critically revised the manuscript; X.Z. participated in data analysis; R.Y. participated in the concept design of the study. All authors gave final approval for publication and agree to be held accountable for the work performed therein.

Competing interests. We declare we have no competing interests.

Funding. This work was supported by the National Natural Science Foundation of China (grant nos. 563 16Z103060007).

Acknowledgements. The authors thank Dr Ping Ao for his help in the design of this study. The authors thank Dr Yong-cong Chen and Dr Lijian Hui for their help in writing.

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
