## [Peer Review File · Royal Society Open Science]

Review History

RSOS-202004.R0 (Original submission)

Review form: Reviewer 1

Is the manuscript scientifically sound in its present form?

Yes

Are the interpretations and conclusions justified by the results?

Yes

Is the language acceptable?

Yes

Do you have any ethical concerns with this paper?

No

Have you any concerns about statistical analyses in this paper?

No

Recommendation?

Accept with minor revision (please list in comments)

Comments to the Author(s)

The study presented in the manuscript titled “The complex landscape of hematopoietic lineage commitments is encoded in the coarse-grained endogenous network” is related to the problem of hematopoietic cell differentiation. The authors show that the process of hematopoietic lineage commitment can be reproduced based on the landscape formed by the dynamics of the core-level regulatory network. The network was obtained by considering the key transcription factors that determine lineage specifications at each cell-fate branchpoint, i.e. genetic switch-like gene pairs working antagonistically, and their interactions. The network was constructed by literature mining and the integration of pre-existing biological knowledge. The initial model was refined by considering the discrepancies between the model behaviour and the experimental results. In particular, the addition of autoregulation of IKAROS was essential to cover known cell types by the model. This modification was justified by the investigation of ChIP-Seq data. The final model was capable of reproducing the behaviour in agreement with wet-lab experiments. Moreover, it allowed to predict non-trivial characteristics of the differentiation dynamics. In particular, by mapping steady states to the differentiation landscape, not only well-known cell types but also cell states that had not yet been recognised in the current hierarchy were identified. Cell states MPP1 and MPP2 were predicted by the model and verified by scRNA-seq data. Furthermore, the model explained two distinct development routes of megakaryocyte-erythroid lineage. By performing Least Action Path analysis, the fluctuations of transcription factors in these two routes of state transitions were predicted and it was shown that GATA1/PU.1 and GATA1/IKAROS play a crucial role at the initial phase of MPP differentiation.

I find this work relevant, interesting, and sound. In my opinion, the obtained results are important and well-justified. The work is clearly presented in general. There are however certain issues that should be addressed by the authors before the manuscript can be accepted for publication. The detailed concerns and issues are listed below.

Detailed comments

1. Line 40: quantified -> quantify
2. Line 53: “... lambda has been well documented ...” -> “... lambda has documented well ...”
3. Line 149: don't -> do not
4. Lines 150-151: For the statement “... hence we limited their co-expression in our model mathematically.”, please explain in more details how the limitation was introduced.
5. Figure 1 B: Please increase the size of the image.
6. Figure 1, caption: Please add a description of what the blue arrows and the boxes on them represent and explain in the text how they were obtained.
7. Line 175: Based on the description in the Methods section, I understand that “one positive real parts” means that among all eigenvalues there is only one with a positive real part. However, this wording is not very clear and a better explanation should be provided already here.
8. Line 178: Please explain the parameter value choice for “ $n = 4$, $a = 16$ ”.

9. Line 187: For the statement “We obtained equilibrium states”, please explain briefly how the equilibrium states were obtained and please point to the corresponding detailed explanation in the Methods section.
10. Line 189, “Moreover, we used a Boolean logical ...”: Please provide an explanation on how the Boolean model was constructed and how the stable states were obtained. Also, how were they mapped to the stable states of the ODE model. Some of the explanations are already provided in the Methods section. Please refer to them here.
11. Line 191-192, “In the Boolean logical model, all the stable states obtained from the ODE model were reproduced, ...”: Where there any additional stable states in the Boolean model, i.e. ones which had no counterparts in the ODE model? If yes, please list all the states in Figure S5 and provide a discussion on these states.
12. Line 213: “was showing” -> “is shown”
13. Line 215, “B cell were obtained in the network as stable states S10-S15 (Figure 3A).”: States S14-S15 not shown in Fig. 3A. Please add them.
14. Line 265, “We first analyzed the scRNA-Seq data of Lin-CD34+CD38- cells”: Please describe the performed analysis in more details. In particular, how the scRNA-seq expressions were mapped to the expression patterns of model states. Does the mapping of an expression profile to an expression pattern mean that the profile is closest (in what sense?) to the pattern when compared with the distance to all other identified patterns?
15. Lines 268-269: Please briefly explain what the surface phenotype is. Please explain why HSCs/MPPs are indistinguishable in your model.
16. Figure 4, caption: For (B), please mention the compartment CD38+.
17. Line 295: “MPP1 states” -> “MPP1 state”
18. Line 303: Please provide a short, intuitive description of the least action path method.
19. Lines 309-310, “... that LMPPs separating with myeloid lineages instructed dominantly by IKAROS was the first step of MPP differentiation, ...”: This part of the sentence does not read very well. Please rephrase to make it clearer.
20. Line 315: “MPP1 states” -> “MPP1 state”
21. Line 417: “... model and capture quantitative ...” -> “... model and to capture quantitative ...”
22. Line 434: Please add a rationale for “Here we assumed $a = n^2$ ”.
23. Line 435: Please point directly to the right table in the supplementary material.
24. Line 441: “(or called a saddle)” -> “(a.k.a a saddle)”
25. Line 442: “... of its Jacobian ...” -> “... of the Jacobian ...”
26. Line 442: “eigenvalue of” -> “eigenvalue with”
27. Line 443: parts -> part

28. Line 443: “eigenvalue of” -> “eigenvalue with”

29. Line 444: parts -> part

30. Line 445: Please explain in detail how the initial points were sampled. In particular, how did you assure that the space of possible initial conditions is well-covered by the sampling and that all/almost all stable points are identified with your approach.

31. Lines 446-447: Please justify the statement “... can cover attractors owning a relatively large basin in the state space, which means the attractors are always biologically stable.”. How did you check that the attractors have relatively large basins? Please provide more details.

32. Lines 447-449: The sentence is not clear. Please explain in more details what computations were performed and how the conclusion on attractor coverage was drawn.

33. Line 452, “The detailed algorithm has been reported in [27]”: For the study to be self-contained, please briefly present the algorithm and its main assumptions.

34. Line 457, the description of the Boolean model: Please explain what update mode is used for the Boolean model, i.e. synchronous or asynchronous. Please provide a rationale behind your choice. For example, if the update mode is synchronous, which I guess is the case in this study, it is known to introduce spurious attractors. How is this taken care of?

35. Lines 460-461, “... a discrete variable ‘0’ or ‘1’ of each gene represents its expression level low or high expression of a state”: 0 or 1 is a value that the discrete variables can take. Please rephrase this sentence to make it more precise.

36. Line 464: a_{ij} -> r_{ij}

37. Line 467, “Here we generated 106 random vectors ...”: The network is rather small as it consists of 10 nodes. The state space is of size 2^{10} , which is 1024. To identify all the attractors, it is enough to consider all the 1024 distinct states as initial states.

There are also methods based on SAT-solving or decomposition-based methods that can be used for exact computation of all attractors of synchronous Boolean models. Dedicated decomposition-based methods can handle asynchronous Boolean networks.

38. Line 469, description of single-cell expression data analysis: Please provide more details. Please describe what kind of normalisation was used.

39. Line 472, description of least action path analysis: Please provide a short, intuitive explanation of what least action path analysis is and how the algorithm works.

40. Line 474: Please briefly explain what T and N are.

41. Supplementary Figure S2, caption: “... patterns that defined as a monocyte state ...” -> “... patterns that correspond to a monocyte state ...”

Review form: Reviewer 2

Is the manuscript scientifically sound in its present form?

No

Are the interpretations and conclusions justified by the results?

No

Is the language acceptable?

No

Do you have any ethical concerns with this paper?

No

Have you any concerns about statistical analyses in this paper?

No

Recommendation?

Reject

Comments to the Author(s)

General Comments:

This is a very difficult and hard to follow manuscript, and it is not at all clear that the authors have well defined what is new and significantly useful formation.

Specific Comments:

- 1) You use a lot of abbreviations, most of which will not be helpful to the vast readership of this journal. You take much too much for granted, regarding the knowledge of the readership.
- 2) This is a long and complicated read, with numerous references, many which are missing either the Journal, Volume, and/or page numbers. What exactly is the new information presented? You need to present what is new information that would make this a worthwhile addition to the literature. You can greatly decrease parts of the paper and then add information that succinctly gets to the new information in a way that the readers will understand.
- 3) What is: "Coarse Grained Endogenous Network", and how does this differ from other endogenous networks, and why is this important?
- 4) You leave too much for granted regarding the readership of this journal.

Decision letter (RSOS-202004.R0)

Dear Ms Wang

The Editors assigned to your paper RSOS-202004 "The complex landscape of hematopoietic lineage commitments is encoded in the coarse-grained endogenous network" have now received comments from reviewers and would like you to revise the paper in accordance with the reviewer comments and any comments from the Editors. Please note this decision does not guarantee eventual acceptance.

Please submit your revised manuscript and required files (see below) no later than 21 days from today's (ie 11-Mar-2021) date. Note: the ScholarOne system will 'lock' if submission of the revision is attempted 21 or more days after the deadline. If you do not think you will be able to meet this deadline please contact the editorial office immediately.

on behalf of Professor Ion Petre (Associate Editor) and Pietro Cicuta (Subject Editor)
openscience@royalsociety.org

Editor's Comments to the Author:

One reviewer has very detailed feedback, and overall positive impression. The other reviewer has found the paper very difficult to read and appreciate. The authors should address the detailed points of the first reviewer, but also attempt to improve the introduction, discussion and conclusion so that the manuscript can be better appreciated by non-specialists.

Reviewer comments to Author:

Reviewer: 1

Comments to the Author(s)

The study presented in the manuscript titled "The complex landscape of hematopoietic lineage commitments is encoded in the coarse-grained endogenous network" is related to the problem of

hematopoietic cell differentiation. The authors show that the process of hematopoietic lineage commitment can be reproduced based on the landscape formed by the dynamics of the core-level regulatory network. The network was obtained by considering the key transcription factors that determine lineage specifications at each cell-fate branchpoint, i.e. genetic switch-like gene pairs working antagonistically, and their interactions. The network was constructed by literature mining and the integration of pre-existing biological knowledge. The initial model was refined by considering the discrepancies between the model behaviour and the experimental results. In particular, the addition of autoregulation of IKAROS was essential to cover known cell types by the model. This modification was justified by the investigation of ChIP-Seq data. The final model was capable of reproducing the behaviour in agreement with wet-lab experiments. Moreover, it allowed to predict non-trivial characteristics of the differentiation dynamics. In particular, by mapping steady states to the differentiation landscape, not only well-known cell types but also cell states that had not yet been recognised in the current hierarchy were identified. Cell states MPP1 and MPP2 were predicted by the model and verified by scRNA-seq data. Furthermore, the model explained two distinct development routes of megakaryocyte-erythroid lineage. By performing Least Action Path analysis, the fluctuations of transcription factors in these two routes of state transitions were predicted and it was shown that GATA1/PU.1 and GATA1/IKAROS play a crucial role at the initial phase of MPP differentiation.

I find this work relevant, interesting, and sound. In my opinion, the obtained results are important and well-justified. The work is clearly presented in general. There are however certain issues that should be addressed by the authors before the manuscript can be accepted for publication. The detailed concerns and issues are listed below.

Detailed comments

1. Line 40: quantified -> quantify
2. Line 53: "... lambda has been well documented ..." -> "... lambda has documented well ..."
3. Line 149: don't -> do not
4. Lines 150-151: For the statement "... hence we limited their co-expression in our model mathematically.", please explain in more details how the limitation was introduced.
5. Figure 1 B: Please increase the size of the image.
6. Figure 1, caption: Please add a description of what the blue arrows and the boxes on them represent and explain in the text how they were obtained.
7. Line 175: Based on the description in the Methods section, I understand that "one positive real parts" means that among all eigenvalues there is only one with a positive real part. However, this wording is not very clear and a better explanation should be provided already here.
8. Line 178: Please explain the parameter value choice for " $n = 4, a = 16$ ".
9. Line 187: For the statement "We obtained equilibrium states", please explain briefly how the equilibrium states were obtained and please point to the corresponding detailed explanation in the Methods section.
10. Line 189, "Moreover, we used a Boolean logical ...": Please provide an explanation on how the Boolean model was constructed and how the stable states were obtained. Also, how were they

mapped to the stable states of the ODE model. Some of the explanations are already provided in the Methods section. Please refer to them here.

11. Line 191-192, "In the Boolean logical model, all the stable states obtained from the ODE model were reproduced, ...": Where there any additional stable states in the Boolean model, i.e. ones which had no counterparts in the ODE model? If yes, please list all the states in Figure S5 and provide a discussion on these states.

12. Line 213: "was showing" -> "is shown"

13. Line 215, "B cell were obtained in the network as stable states S10-S15 (Figure 3A).": States S14-S15 not shown in Fig. 3A. Please add them.

14. Line 265, "We first analyzed the scRNA-Seq data of Lin-CD34+CD38- cells": Please describe the performed analysis in more details. In particular, how the scRNA-seq expressions were mapped to the expression patterns of model states. Does the mapping of an expression profile to an expression pattern mean that the profile is closest (in what sense?) to the pattern when compared with the distance to all other identified patterns?

15. Lines 268-269: Please briefly explain what the surface phenotype is. Please explain why HSCs/MPPs are indistinguishable in your model.

16. Figure 4, caption: For (B), please mention the compartment CD38+.

17. Line 295: "MPP1 states" -> "MPP1 state"

18. Line 303: Please provide a short, intuitive description of the least action path method.

19. Lines 309-310, "... that LMPPs separating with myeloid lineages instructed dominantly by IKAROS was the first step of MPP differentiation, ...": This part of the sentence does not read very well. Please rephrase to make it clearer.

20. Line 315: "MPP1 states" -> "MPP1 state"

21. Line 417: "... model and capture quantitative ..." -> "... model and to capture quantitative ..."

22. Line 434: Please add a rationale for "Here we assumed $a = n^2$ ".

23. Line 435: Please point directly to the right table in the supplementary material.

24. Line 441: "(or called a saddle)" -> "(a.k.a a saddle)"

25. Line 442: "... of its Jacobian ..." -> "... of the Jacobian ..."

26. Line 442: "eigenvalue of" -> "eigenvalue with"

27. Line 443: parts -> part

28. Line 443: "eigenvalue of" -> "eigenvalue with"

29. Line 444: parts -> part

30. Line 445: Please explain in detail how the initial points were sampled. In particular, how did you assure that the space of possible initial conditions is well-covered by the sampling and that all/almost all stable points are identified with your approach.
31. Lines 446-447: Please justify the statement "... can cover attractors owning a relatively large basin in the state space, which means the attractors are always biologically stable.". How did you check that the attractors have relatively large basins? Please provide more details.
32. Lines 447-449: The sentence is not clear. Please explain in more details what computations were performed and how the conclusion on attractor coverage was drawn.
33. Line 452, "The detailed algorithm has been reported in [27]": For the study to be self-contained, please briefly present the algorithm and its main assumptions.
34. Line 457, the description of the Boolean model: Please explain what update mode is used for the Boolean model, i.e. synchronous or asynchronous. Please provide a rationale behind your choice. For example, if the update mode is synchronous, which I guess is the case in this study, it is known to introduce spurious attractors. How is this taken care of?
35. Lines 460-461, "... a discrete variable '0' or '1' of each gene represents its expression level low or high expression of a state": 0 or 1 is a value that the discrete variables can take. Please rephrase this sentence to make it more precise.
36. Line 464: $a_{\{ij\}} \rightarrow r_{\{ij\}}$
37. Line 467, "Here we generated 106 random vectors ...": The network is rather small as it consists of 10 nodes. The state space is of size 2^{10} , which is 1024. To identify all the attractors, it is enough to consider all the 1024 distinct states as initial states.
- There are also methods based on SAT-solving or decomposition-based methods that can be used for exact computation of all attractors of synchronous Boolean models. Dedicated decomposition-based methods can handle asynchronous Boolean networks.
38. Line 469, description of single-cell expression data analysis: Please provide more details. Please describe what kind of normalisation was used.
39. Line 472, description of least action path analysis: Please provide a short, intuitive explanation of what least action path analysis is and how the algorithm works.
40. Line 474: Please briefly explain what T and N are.
41. Supplementary Figure S2, caption: "... patterns that defined as a monocyte state ..." \rightarrow "... patterns that correspond to a monocyte state ..."

Reviewer: 2

Comments to the Author(s)

General Comments:

This is a very difficult and hard to follow manuscript, and it is not at all clear that the authors have well defined what is new and significantly useful formation.

Specific Comments:

- 1) You use a lot of abbreviations, most of which will not be helpful to the vast readership of this journal. You take much too much for granted, regarding the knowledge of the readership.
- 2) This is a long and complicated read, with numerous references, many which are missing either the Journal, Volume, and/or page numbers. What exactly is the new information presented? You need to present what is new information that would make this a worthwhile addition to the literature. You can greatly decrease parts of the paper and then add information that succinctly gets to the new information in a way that the readers will understand.
- 3) What is: Coarse Grained Endogenous Network", and how does this differ from other endogenous networks, and why is this important?
- 4) You leave too much for granted regarding the readership of this journal.

===PREPARING YOUR MANUSCRIPT===

===PREPARING YOUR REVISION IN SCHOLARONE===

Author's Response to Decision Letter for (RSOS-202004.R0)

See Appendix A.

Decision letter (RSOS-202004.R1)

Dear Ms Wang

The Editors assigned to your paper RSOS-202004.R1 "The complex landscape of hematopoietic lineage commitments is encoded in the coarse-grained endogenous network" have made a decision based on their reading of the paper and any comments received from reviewers.

Regrettably, in view of the reports received, the manuscript has been rejected in its current form. However, a new manuscript may be submitted which takes into consideration these comments.

We invite you to respond to the comments supplied below and prepare a resubmission of your manuscript. Below the referees' and Editors' comments (where applicable) we provide additional requirements. We provide guidance below to help you prepare your revision.

Please note that resubmitting your manuscript does not guarantee eventual acceptance, and we do not generally allow multiple rounds of revision and resubmission, so we urge you to make every effort to fully address all of the comments at this stage. If deemed necessary by the Editors, your manuscript will be sent back to one or more of the original reviewers for assessment. If the original reviewers are not available, we may invite new reviewers.

Please resubmit your revised manuscript and required files (see below) no later than 20-Oct-2021. Note: the ScholarOne system will 'lock' if resubmission is attempted on or after this deadline. If you do not think you will be able to meet this deadline, please contact the editorial office immediately.

Please note article processing charges apply to papers accepted for publication in Royal Society Open Science (<https://royalsocietypublishing.org/rsos/charges>). Charges will also apply to papers transferred to the journal from other Royal Society Publishing journals, as well as papers submitted as part of our collaboration with the Royal Society of Chemistry (<https://royalsocietypublishing.org/rsos/chemistry>). Fee waivers are available but must be requested when you submit your manuscript (<https://royalsocietypublishing.org/rsos/waivers>).

Thank you for submitting your manuscript to Royal Society Open Science and we look forward to receiving your resubmission. If you have any questions at all, please do not hesitate to get in touch.

on behalf of Professor Ion Petre (Associate Editor) and Pietro Cicuta (Subject Editor)
openscience@royalsociety.org

Associate Editor Comments to Author (Professor Ion Petre):

The manuscript continues to lack in clarity, especially in: its underlying hypothesis, its novelty over the existing literature, the details of the mathematical model and its connection to biological knowledge, the choice of the numerical setup of the model, the validation of the model. The additional discussions on the Boolean network model remains somewhat disconnected from that on the Hill equation model.

===PREPARING YOUR MANUSCRIPT===

===PREPARING YOUR REVISION IN SCHOLARONE===

Please ensure that you include a summary of your paper at Step 2 'Type, Title, & Abstract'. This should be no more than 100 words to explain to a non-scientific audience the key findings of your

research. This will be included in a weekly highlights email circulated by the Royal Society press office to national UK, international, and scientific news outlets to promote your work.

Author's Response to Decision Letter for (RSOS-202004.R1)

See Appendix B.

RSOS-211289.R0

Review form: Reviewer 2

Is the manuscript scientifically sound in its present form?

Yes

Are the interpretations and conclusions justified by the results?

Yes

Is the language acceptable?

Yes

Do you have any ethical concerns with this paper?

No

Have you any concerns about statistical analyses in this paper?

No

Recommendation?

Accept as is

Comments to the Author(s)

None.

Decision letter (RSOS-211289.R0)

Dear Ms Wang,

I am pleased to inform you that your manuscript entitled "The complex landscape of haematopoietic lineage commitments is encoded in the coarse-grained endogenous network" is now accepted for publication in Royal Society Open Science.

on behalf of Professor Ion Petre (Associate Editor) and Pietro Cicuta (Subject Editor)
openscience@royalsociety.org

Reviewer comments to Author:
Reviewer: 2
Comments to the Author(s)
None.

Appendix A

Dear editor and reviewers:

We would like to thank you for your careful reading, helpful comments, and constructive suggestions, which have significantly improved the presentation of our manuscript.

We have carefully considered all comments from the reviewers and revised our manuscript accordingly. In our revisions, 1) we have improved the writing to increase the readability of the manuscript; 2) according to reviewer's suggestions, a lot of details were added to make the manuscript more explicit and 3) the part of Boolean network was improved; 4) we made corrections of the format of the references, typos and other errors.

In the following, we summarize our responses to each comment from the reviewers. We believe that our responses have well addressed all concerns from the reviewers. We hope our revised manuscript can be accepted for publication.

Best regards,

Mengyao Wang

Below the comments of the reviewer are responses point to point:

Responds to the reviewers' comments:

Reviewer: 1

Comments to the Author(s)

The study presented in the manuscript titled “The complex landscape of hematopoietic lineage commitments is encoded in the coarse-grained endogenous network” is related to the problem of hematopoietic cell differentiation. The authors show that the process of hematopoietic lineage commitment can be reproduced based on the landscape formed by the dynamics of the core-level regulatory network. The network was obtained by considering the key transcription factors that determine lineage specifications at each cell-fate branchpoint, i.e. genetic switch-like gene pairs working antagonistically, and their interactions. The network was constructed by literature mining and the integration of pre-existing biological knowledge. The initial model was refined by considering the discrepancies between the model behaviour and the experimental results. In particular, the addition of autoregulation of IKAROS was essential to cover known cell types by the model. This modification was justified by the investigation of ChIP-Seq data. The final model was capable of reproducing the behaviour in agreement with wet-lab experiments. Moreover, it allowed to predict non-trivial characteristics of the differentiation dynamics. In particular, by mapping steady states to the differentiation landscape, not only well-known cell types but also cell states that had not yet been recognised in the current hierarchy were identified. Cell states MPP1 and MPP2

were predicted by the model and verified by scRNA-seq data. Furthermore, the model explained two distinct development routes of megakaryocyte-erythroid lineage. By performing Least Action Path analysis, the fluctuations of transcription factors in these two routes of state transitions were predicted and it was shown that GATA1/PU.1 and GATA1/IKAROS play a crucial role at the initial phase of MPP differentiation.

I find this work relevant, interesting, and sound. In my opinion, the obtained results are important and well-justified. The work is clearly presented in general. There are however certain issues that should be addressed by the authors before the manuscript can be accepted for publication. The detailed concerns and issues are listed below.

Response: We thank the reviewer for reading our paper carefully and giving the above positive comments. We gratefully thank for the precious time the reviewer spent making such detailed suggestion and constructive remarks, which helped us significantly improve the manuscript. Below the comments of the reviewer are responses point to point and the revisions are indicated.

Detailed comments

1. Line 40: quantified -> quantify

Sorry for our incorrect writing and it has been revised.

2. Line 53: "... lambda has been well documented ..." -> "... lambda has documented well ..."

Sorry for our incorrect writing and it has been revised.

3. Line 149: don't -> do not

Sorry for our incorrect writing and it has been revised.

4. Lines 150-151: For the statement "... hence we limited their co-expression in our model mathematically.", please explain in more details how the limitation was introduced.

It has been revised as "GATA1 and C/EBP α do not interact directly with each other, but they have a competing effect at a functional level. Hence we assumed a negative regulation of GATA1 in the equation describing the evolution of C/EBP α , which then effectively limits their co-expression."

5. Figure 1 B: Please increase the size of the image.

Sorry for our negligence and it has been revised.

6. Figure 1, caption: Please add a description of what the blue arrows and the boxes on them represent and explain in the text how they were obtained.

The description has been added in the caption that “The arrow with boxes on the top of each sub-image presents gene structure, arrow indicates the direction of transcription, and the boxes indicate exons. The information of genes location and structure were obtained from Ensemble dataset.”

7. Line 175: Based on the description in the Methods section, I understand that “one positive real parts” means that among all eigenvalues there is only one with a positive real part. However, this wording is not very clear and a better explanation should be provided already here.

We have modified the wording in the manuscript to avoid misinterpretation. The classification of states in the system is detailed in the Methods section: The equilibrium points are the points x which satisfied $\frac{dx}{dt} = 0$. The equilibrium points of the system can be categorized by the signs of the eigenvalues of their Jacobian matrix. If the real parts of the eigenvalues are all negative, the equilibrium point is identified as a stable state. If one or more eigenvalues have a positive real part, the equilibrium point is an unstable state (a.k.a. a saddle point). With respect to the number of the eigenvalues of the Jacobian matrix having a positive real part, we defined an unstable state with a single eigenvalue of positive real part as a transition state, and those with two or more eigenvalues of positive real part as hyper-transition states.

8. Line 178: Please explain the parameter value choice for “n = 4, a = 16”.

We obtained equilibrium states in our ODE model with different Hill coefficients and all equilibrium states were conserved only with subtle alterations in values. That signifies the robustness of this dynamical model under different parameters, and exact parameters are not important. Given such robustness, there is no particular reason for choosing this set of parameters to presented and exerted subsequent analysis. We have modified the manuscript on this issue.

9. Line 187: For the statement “We obtained equilibrium states”, please explain briefly how the equilibrium states were obtained and please point to the corresponding detailed explanation in the Methods section.

It has been described in the manuscript as “Mathematically, they are equilibrium states, namely the points x which satisfied $\frac{dx}{dt} = 0$ of the ordinary differential equations (ODEs) for the network dynamics.”. The details could be seen in Method, ODE model section.

10. Line 189, “Moreover, we used a Boolean logical ...”: Please provide an explanation on how the Boolean model was constructed and how the stable states were obtained. Also, how were they mapped to the stable states of the ODE model. Some of the explanations are already provided in the Methods section. Please refer to them here.

The details of the construction of Boolean model and how stable states were obtained have been described in Method, Boolean dynamics model section, and it has been noted. As for mapping to the stable states of the ODE model, in the revised manuscript we explained this issue as “In the Boolean logical model, all the stable states obtained from the ODE model were reproduced. That is to say, we compared the stable states obtained in the Boolean model with the stable states obtained in the ODE, and found there is a one-to-one correspondence between them, in the sense of the levels (high/low) of each elements of every stable state (Figure 2B and Figure S5).”

11. Line 191-192, “In the Boolean logical model, all the stable states obtained from the ODE model were reproduced, ...”: Where there any additional stable states in the Boolean model, i.e. ones which had no counterparts in the ODE model? If yes, please list all the states in Figure S5 and provide a discussion on these states.

No, there is no additional stable state in the Boolean model. All the states obtained in the model have been listed in Figure S5.

12. Line 213: “was showing” -> “is shown”

Sorry for our incorrect writing and it has been revised.

13. Line 215, “B cell were obtained in the network as stable states S10-S15 (Figure 3A).”: States S14-S15 not shown in Fig. 3A. Please add them.

Sorry for this mistake, the correct description is “The states representing unipotent progenitors of erythrocyte, megakaryocyte, neutrophil, monocyte, and B cell were obtained in the network as stable states S9-S13 (Figure 3A)”, and it has been revised in the manuscript.

14. Line 265, “We first analyzed the scRNA-Seq data of Lin-CD34+CD38- cells”: Please describe the performed analysis in more details. In particular, how the scRNA-seq expressions were mapped to the expression patterns of model states. Does the mapping of an expression profile to an expression pattern mean that the profile is closest (in what sense?) to the pattern when compared with the distance to all other identified patterns?

We first analyzed the scRNA-Seq data of Lin-CD34+CD38- cells, examining the expression profile of the ten core transcription factors as well as other fourteen lineage specific genes (Figure 4A). Specifically, in the profile of Lin-CD34+CD38- cells, we checked whether there are cells with the specific expression pattern that corresponds to each stable state we obtained.

For example, cells with expression pattern, that the $\text{Log}_2(\text{counts} + 1)$ of each gene

according with $(\text{RUNX1} > 0 \ \& \ \text{SPI1} \geq 0 \ \& \ \text{CEBPA} > 0 \ \& \ \text{GFI1} \geq 0 \ \& \ \text{EGR1} \geq 0 \ \& \ \text{GATA1} \geq 0 \ \& \ \text{KLF1} = 0 \ \& \ \text{FLI1} > 0 \ \& \ \text{IKZF1} > 0 \ \& \ \text{EBF1} = 0)$ were mapped to MPP state(T1).

More details have been added in the Methods, Single-cell expression data analysis section.

15. Lines 268-269: Please briefly explain what the surface phenotype is. Please explain why HSCs/MPPs are indistinguishable in your model.

Cells with the expression pattern we defined as MPP were also located in Lin-CD34+CD38-CD45R-CD90+ compartment partially, which is the surface markers applied to isolated HSCs. As can be seen from Figure 4A, cells with MPP and HSC markers (the phenotype bar in the figure corresponds to dark purple and blue respectively) were mapped to our defined MPP state, indicating that HSCs/MPPs were almost indistinguishable from this model.

16. Figure 4, caption: For (B), please mention the compartment CD38+.

It has been revised as “(B) Verification of the predicted more differentiated progenitors which is marked by CD38+ including CMP-like, MEP-like, GMP-like, and CLP-like states in scRNA-Seq data.”

17. Line 295: “MPP1 states” -> “MPP1 state”

Sorry for our incorrect writing and it has been revised.

18. Line 303: Please provide a short, intuitive description of the least action path method.

Transitions among stable states are triggered by noises. For cell state transitions, the noise could be transcriptional noises or fluctuating signals. Transitions are probabilistic in the stochastically perturbed dynamical system. The least action path (LAP) method provides an algorithm to obtain the most probable transition by minimizing the action along the transition path. The action appears in an exponential decaying factor associated with the probability of a transition.

19. Lines 309-310, “... that LMPPs separating with myeloid lineages instructed dominantly by IKAROS was the first step of MPP differentiation, ...”: This part of the sentence does not read very well. Please rephrase to make it clearer.

It has been revised as “The result implied that in this route LMPPs which separate the myeloid lineages are predominantly instructed by IKAROS. The separation was the first step of MPP differentiation. Thereafter, the antagonism of GATA1 and PU.1 directed further cell-fate choices in the CMP context. ”

20. Line 315: “MPP1 states” -> “MPP1 state”

Sorry for our incorrect writing and it has been revised.

21. Line 417: “... model and capture quantitative ...” -> “... model and to capture quantitative ...”

Sorry for our incorrect writing and it has been revised.

22. Line 434: Please add a rationale for “Here we assumed $a = n^2$.”

First, sorry for our incorrect writing, and it should be “Here we assumed $a = 2^n$ ”. The rationale for this assumption is:

Even if activation and inhibition can be described in terms of the Hill equation, there are a large number of parameters involved in the equation that cannot be obtained. In order to select a reasonable way to solve the problem, we use the dimensionless quantitative description framework of the relative expression level of proteins, which is normalized between ‘0’ and ‘1’. ‘0’ represents the lowest level and ‘1’ represents the highest level. The maximum protein production rate is V_{x_i} , and it is normalized to ‘1’. According to the form of the equation that

we used, the rate of production of x_i is $f([\text{activator}]) = V_{x_i} \cdot \frac{a \cdot [\text{activator}]^n}{1 + a \cdot [\text{activator}]^n}$. In

order to reasonably describe the activation and inhibition under the normalization,

when $[\text{activator}] = \frac{1}{2}$, the rate of production of x_i should be $f([\text{activator}]) = \frac{1}{2}$. Thus

it is concluded that $a = 2^n$.

23. Line 435: Please point directly to the right table in the supplementary material.

All of the ten equations were listed in the supplementary material (Table S1).

24. Line 441: “(or called a saddle)” -> “(a.k.a a saddle)”

Sorry for our incorrect writing and it has been revised.

25. Line 442: “... of its Jacobian ...” -> “... of the Jacobian ...”

Sorry for our incorrect writing and it has been revised.

26. Line 442: “eigenvalue of” -> “eigenvalue with”

Sorry for our incorrect writing and it has been revised.

27. Line 443: parts -> part

Sorry for our incorrect writing and it has been revised.

28. Line 443: “eigenvalue of” -> “eigenvalue with”

Sorry for our incorrect writing and it has been revised.

29. Line 444: parts -> part

Sorry for our incorrect writing and it has been revised.

30. Line 445: Please explain in detail how the initial points were sampled. In particular, how did you assure that the space of possible initial conditions is well-covered by the sampling and that all/almost all stable points are identified with your approach.

A large number of random initial points were performed numerical computations, and Newton iterative method is used to solve the equations, to obtain the equilibrium point in the system. The random vector is made up of random numbers which uniformly distributed in the interval (0,1), generated by MATLAB function 'rand', and the vector dimension is the number of equations. As the number of simulations increases, the results tend to convergent, which means the number of stable states no longer increases. And this explanation has been revised in the manuscript.

31. Lines 446-447: Please justify the statement "... can cover attractors owning a relatively large basin in the state space, which means the attractors are always biologically stable.". How did you check that the attractors have relatively large basins? Please provide more details.

If there is a stable state in the system with a relatively large attraction basin (meaning a large number of random initial points will be 'attracted to' this state), it can be obtained under such large-scale random simulations. The model serves as the reflection of a biological system, so we consider that mathematically steady states are biologically significant. And this explanation has been revised in the manuscript.

32. Lines 447-449: The sentence is not clear. Please explain in more details what computations were performed and how the conclusion on attractor coverage was drawn.

As has been noted, random initial points were performed numerical computations to obtain attractors in the system. As the number of simulations increases, the results tend to convergent, which means the number of obtained stable states no longer increases. Here we used 10^5 random initial points, 10^6 random initial points, and 10^8 random initial points for numerical computations and we got the same number of attractors. That indicated 10^5 computations are sufficient to obtain attractors. And this explanation has been revised in the manuscript.

33. Line 452, "The detailed algorithm has been reported in [27]": For the study to be self-contained, please briefly present the algorithm and its main assumptions.

A brief description of the algorithm has been added in the revised manuscript: The least action path (LAP) method [80] calculates the most probable transition path by minimizing the energy cost, that is the least action, along with the state transition under the stochastic dynamical system. The details has been added in the Methods section.

**34. Line 457, the description of the Boolean model: Please explain what update mode is used for the Boolean model, i.e. synchronous or asynchronous. Please provide a rationale behind your choice. For example, if the update mode is synchronous, which I guess is the case in this study, it is known to introduce spurious attractors. How is this taken care of?**

The update mode for the Boolean network is synchronous. Although such a mode is known to ocassionally introduce spurious attractors, they are not present in the our model. The number of nodes are limited and the relationship between them are carefully examined to avoid the problem.

**35. Lines 460-461, "… a discrete variable '0' or '1' of each gene represents its expression level low or high expression of a state": 0 or 1 is a value that the discrete variables can take. Please rephrase this sentence to make it more precise.**

In the revised manuscript, the sentence was rephrased as "In the model, each node $t$ has only two states, $S(t)=1$ and $S(t)=0$, representing the high expression level state and the low expression level state."

**36. Line 464: a_{ij} -> r_{ij}**

Sorry for our incorrect writing and it has been revised.

**37. Line 467, "Here we generated 106 random vectors …": The network is rather small as it consists of 10 nodes. The state space is of size 2^10, which is 1024. To identify all the attractors, it is enough to consider all the 1024 distinct states as initial states.**

**There are also methods based on SAT-solving or decomposition-based methods that can be used for exact computation of all attractors of synchronous Boolean models. Dedicated decomposition-based methods can handle asynchronous Boolean networks.**

Thanks for the suggestion. We improved this part of the work. First, we study the attractors of the network dynamics by starting from each of the 2^10 = 1024 initial states in the 10-node network (Figure A). We recorded the attractor that each initial state reaches(Figure B). We find that all of the initial states, if they can reach an attractor, eventually flow into one of the 15 stationary states (attractors) shown in Figure C.

Figure A. All of the possible initial states ($2^{10} = 1024$) in the Boolean network. Each column represents one initial state. The value '1' is colored by red, and the value '0' is colored by green.

Figure B. The end states corresponding to the initial states. The value '1' is colored by red, the value '0' is colored by green, and the yellow columns indicate the initial states can not reach an attractor.

	1	2	3	4	5	6	7	8	9	10	11	12	13	14	15	
RUNX1	1	1	1	0	1	0	0	1	0	0	0	0	0	0	0	0
PU.1	0	1	1	1	0	1	1	0	0	0	0	1	0	0	0	0
C/EBP α	1	1	1	1	0	1	0	0	0	0	1	0	0	1	0	0
GFI1	1	0	0	0	0	0	0	0	0	0	1	0	1	1	1	0
EGR1	0	1	1	1	0	1	1	0	0	0	0	1	0	0	0	0
GATA1	0	0	0	0	1	0	0	1	1	1	0	0	0	0	0	0
EKLf	0	0	0	0	0	0	0	1	1	0	0	0	0	0	0	0
FLI1	0	1	1	1	1	1	1	0	0	1	0	1	0	0	0	0
IKAROS	1	1	0	1	0	0	1	0	0	0	0	0	1	1	0	0
EBF1	0	0	0	0	0	0	1	0	0	0	0	0	1	0	0	0

Figure C. All of the non-redundant attractors obtained in Boolean model.

Thanks for your insight into the Boolean model. Since the Boolean model is only a reference in our discussion, we have not further investigated this. We are very interested in this and consider using the Boolean model for the main research in our next work.

38. Line 469, description of single-cell expression data analysis: Please provide more details. Please describe what kind of normalisation was used.

This section has been modified in the revised manuscript as “The single-cell RNA-Seq data of Lin-CD34+ bone marrow cells were used to validate the model predictions in Figure 4. We used the value $\text{Log}_2(\text{counts} + 1)$ of each gene as their expression levels, and the counts of genes were obtained from GSE75478.

A visualized map of expression levels of the ten transcription factors in different lineages is shown in Figure S6. This t-distributed stochastic neighbor embedding (t-SNE) presentation of sorted Lin-CD34+ bone marrow cells was performed by R package Seurat[99][100]. The input data were counts of single-cell RNA-Seq of genes from GSE117498. The normalization method was ‘LogNormalize’ in Seurat.”

39. Line 472, description of least action path analysis: Please provide a short, intuitive explanation of what least action path analysis is and how the algorithm works.

40. Line 474: Please briefly explain what T and N are.

39,40: This section has been modified in the revised manuscript. The least action path (LAP) method [89] calculates the most probable transition path by minimizing the energy cost, that is the least action, along with the state transition under the stochastic dynamical system. The least action path is x which minimizes the action function $S(x)$, which is given by

$$S_{T_1 T_2}(x) = \frac{1}{4} \int_{T_1}^{T_2} \langle \dot{x} - f(x), D^{-1} [\dot{x} - f(x)] \rangle ds$$

and it was discretized as

$$S_{T_1 T_2}(x(t)) = \frac{1}{4} \Delta t \sum_{k=1}^N \sum_{i=1}^M \left\| \frac{x_i^{k+1} - x_i^k}{\Delta t} - \frac{f_i^{k+1} + f_i^k}{2} \right\|^2$$

Given the time interval $[T_1, T_2]$ of the trajectory, and it was divided into N equal subintervals,

$$T_1 = t_1 < t_2 < \dots < t_{N+1} = T_2$$

The minima of the action functional are calculated by MATLAB function `fminunc`. The line segments connecting the initial states and terminal states are used as initial paths. $T = 10$ and $N = 200$ were used. Testing larger T and N , the results were of convergence.

41. Supplementary Figure S2, caption: "... patterns that defined as a monocyte state ..."
-> "... patterns that correspond to a monocyte state ..."

Sorry for our incorrect writing and it has been revised.

Reviewer: 2

Comments to the Author(s)

General Comments:

This is a very difficult and hard to follow manuscript, and it is not at all clear that the authors have well defined what is new and significantly useful formation.

Response: Thanks for the comments. We have revised the article as a whole to increase the readability of the manuscript and attempt to make it can be better appreciated.

Specific Comments:

1) You use a lot of abbreviations, most of which will not be helpful to the vast readership of this journal. You take much to much for granted, regarding the knowledge of the readership.

We feel sorry for the inconvenience brought to the reviewer and readers. We have added a list of abbreviations into the supplementary material (Supplementary Table S2) to help with reading.

2) This is a long and complicated read, with numerous references, many which are missing either the Journal, Volume, and/or page numbers. What exactly is the new information presented? You need to present what is new information that would make this a worthwhile addition to the literature. You can greatly decrease parts of the paper and then add information that succinctly gets to the new information in a way that the readers will understand.

Thanks for the comments. We have rechecked the format of the references and made corrections. We have improved the writing according to your suggestions.

3) What is: Coarse Grained Endogenous Network", and how does this differ from other endogenous networks, and why is this important?

Thank you for raising the issue that we missed in the original manuscript, and we have added the explanation to the discussion section. On two levels, we considered the network as a coarse-grained endogenous network: firstly, a coarse-grained quantitative framework was used to describe activation and inhibition based on the Hill equation. Under this framework, the specific regulatory mechanism among proteins at the micro-level and parameters of biochemical reactions can be ignored. Only a reasonable equation form is applied to describe the activation or inhibition relationship between proteins. Secondly, only most core regulators involving hematopoietic lineage commitments were used to model this complex process. Such a limited number of regulators serves as a coarse-grained description of this process. Considering the accessibility of real regulation mechanisms, biochemical reaction parameters, and the complexity of calculation, this coarse-grained quantitative framework is important and indispensable.

4) You leave too much for granted regarding the readership of this journal.

Thanks for your comments. We have followed the editor's suggestion and have improved the introduction, discussion and conclusion. Hopefully the manuscript can now be better appreciated by non-specialists.

Appendix B

Dear editors:

We would like to thank you for the time and effort spent reviewing the manuscript. According to your helpful comments, we have significantly improved the presentation of our manuscript.

We have carefully considered all the points in the decision letter and all parts pointed out to be unclear are rewritten in this revised version. In our revision, according to the editor's comments, 1) a lot of details were added to make the manuscript more explicit; especially, the underlying assumptions or hypothesis in the section *Introduction* and the section *Discussion*, and the modelling implementation in the section *Methods*; 2) the part of the Boolean method was improved in the manuscript, and more details of results and its connection with Hill equation model were discussed in the *electronic supplementary material file 1, Section: Boolean logic rules*; 3) the part of validation the model were improved, and all the information of the data used to validate the model has been organized into a file and uploaded as the *electronic supplementary material file 2* for a clearer presentation; 4) we have improved the writing to increase the readability of the manuscript. And we have made corrections to the format of the references, typos, and other errors.

We tried our best to improve the clarity of the manuscript. These changes will not influence the results and framework of the paper. Hopefully the manuscript can now be better appreciated by readers. We appreciate for editors and reviewers' warm work earnestly and hope that this version will meet with approval.

Best regards,

Mengyao Wang

In the following, we summarize our responses to each point from the editors' comments:

We have improved the clarity of the whole manuscript, especially in the following parts you pointed out.

1. its underlying hypothesis

We employed the endogenous molecular-cellular network modelling framework [1,2], which has been tested to be effective in previous studies [3–9]. The main hypothesis is that genotype and phenotype can be bridged by network dynamics. Cellular phenotypes correspond to robust states emerging from the dynamics of a gene regulatory network. The idea that regarding attractor states of the gene regulatory network dynamics as cellular phenotypes goes back to Delbrück, Jacob and Monod, and followed by Kauffman [10–12]. This assumption was also demonstrated by some recent studies of gene regulatory networks [13–

15]. Some studies have reviewed and developed such ideas, for example in [16–18]. We restate this point in the manuscript section *Introduction*.

In addition, some hypotheses were adopted to implement the model practically. First, when constructing the network, a minimal-scale network is considered. We focused on genetic switch-like gene pairs, which work antagonistically at each cell-fate branchpoint. This idea tested be effective from our previous study on the Phage λ genetic switch [19]. The endogenous network composed by CI and Cro and their cross-antagonism was found to be able to explain experimental data quantitatively in the choice of alternative survival modes of Phage λ , lytic or lysogenic. Second, after constructing a primary network, we refined the network based on the hypothesis that calculated states of the network dynamics present all expected known cell types. If not, reasonable regulations should be added. Third, the interactions between factors are considered simply as ‘activation’ or ‘inhibition’, both in the graph of the network and in the mathematical model. Considering the inaccessibility of details of regulation mechanisms and biochemical parameters, this coarse-grained implementation is indispensable. We found that our model is robust in parameter variations. Studies of simple parameter-free systems, such as discrete Boolean networks, suggest that the characteristics of network dynamics, such as the presence of attractor states, do not depend much on quantitative details of interaction parameters but rather, on the network topology [20]. The above ideas are presented in the revised manuscript in *Discussion*.

2. its novelty over the existing literature

The novelty and the significance of this work have been recognized by the Reviewer 1. We emphatically addressed this issue in *Discussion*.

3. the details of the mathematical model and its connection to biological knowledge

First, the details of the mathematical model: we have added details and tried our best to explain the modelling approach clearer. We believe the section *Methods* has been improved.

Second, its connection to biological knowledge: we have improved the writing of the results. The original content results 2 and 3 were rewritten. They were reorganized as section “**2.2 From molecular network to cellular phenotype:**” and two subsections “**2.2.1. Robust states and their transition paths**” and “**2.2.2. The well-known cell types in haematopoietic lineage commitments are reproduced, and model predicted novel cell types are revealed from the landscape**”. Combined with the content of the underlying hypothesis, we hope the model’s connection to biological knowledge is clear for readers.

4. the choice of the numerical setup of the model

I am not quite sure what the numerical setup is saying. I suppose it means parameters of ODEs or random numerical simulations.

1. For the parameters of ODEs, 1) the rationale for the constraint $a = 2^n$ was discussed in **Methods 4.2.1 ODEs**: “The parameter n indicates the Hill coefficient of the Hill function, while a indicates the apparent dissociation constant. The value of each factor x_i was normalized from 0 to 1. We can reasonably assume that when $[\text{activator}] = 0$, then $f_{\text{activation}} = 0$; when $[\text{activator}] = 1$; then $f_{\text{activation}} = 1$; when $[\text{activator}] = \frac{1}{2}$, then $f_{\text{activation}} = \frac{1}{2}$. Thus, we can deduce that $a = 2^n$ ”.

2) the reason we tested parameters in the range $n = 3, 4, \dots, 10$, $a = 8, 16, \dots, 1024$, was discussed in the **electronic supplementary material 1.2 Supplementary simulation results iii Robustness under variations of parameters**: “Quantitative studies of signal transduction systems show that usually Hill coefficient $n \geq 3$, such as cell cycle regulation [4], MAPK pathways [4], Ras pathways [5] and Notch signaling pathways [6]. And when $n \geq 10$ the equations tend to be discrete Boolean functions (discussed in next section). Thus, we tested the parameter range $3 \leq n \leq 10$ in the ODEs and Boolean rules to evaluate parametric robustness”. Detailed results of parameter testing were also provided in the **electronic supplementary material iii Robustness under variations of parameters, Supplementary Table S1**.

2. For the random numerical simulations, we restated this part in section **Methods 4.2.1 ODEs: Attractor states and transition states**. “The fixed points are the points x that satisfy $\frac{dx}{dt} = 0$. These points were calculated by the MATLAB function ‘fsolve’, which is a numerical method for solving nonlinear differential equations based on Newton iteration algorithm. When calculating fixed points, an initial point needs to be given to executed iteration algorithm. A large number of random initial points were used to computed fixed points. The random initial point (a vector whose dimension is the number of factors in the network) uniformly distributes in the interval $[0, 1]$, which is generated by the MATLAB function ‘rand’.”

5. the validation of the model

The model was validated by RNA-Seq data. We improve the writing of the sections **Results 2.3 Validation of the modelling results** and **Methods 4.3. Modelling results validation**. And all the information of this section was provided in detail in uploaded **electronic supplementary**

material file 2.

6. The additional discussions on the Boolean network model remains somewhat disconnected from that on the Hill equation model

The connection between Hill equation model and Boolean network model was discussed in the *electronic supplementary material 2.3 Connections between Boolean rules versus ODEs*. In brief, when the Hill coefficient $n \longrightarrow +\infty$, the Hill equation be equivalent to the Boolean rules, and it was shown in **Figure S8**

Reference

1. Ao P, Galas D, Hood L, Zhu X. 2008 Cancer as robust intrinsic state of endogenous molecular-cellular network shaped by evolution. *Med. Hypotheses* **70**, 678–684. (doi:10.1016/j.mehy.2007.03.043)
2. Yuan R, Zhu X, Wang G, Li S, Ao P. 2017 Cancer as robust intrinsic state shaped by evolution: a key issues review. *Reports Prog. Phys.* **80**, 042701. (doi:10.1088/1361-6633/aa538e)
3. Wang G, Zhu X, Gu J, Ao P. 2014 Quantitative implementation of the endogenous molecular–cellular network hypothesis in hepatocellular carcinoma. *Interface Focus* **4**, 20130064. (doi:10.1098/rsfs.2013.0064)
4. Li S, Zhu X, Liu B, Wang G, Ao P. 2015 Endogenous molecular network reveals two mechanisms of heterogeneity within gastric cancer. *Oncotarget* **6**, 13607–13627. (doi:10.18632/oncotarget.3633)
5. Yuan R *et al.* 2017 Beyond cancer genes: colorectal cancer as robust intrinsic states formed by molecular interactions. *Open Biol.* **7**, 170169. (doi:10.1098/rsob.170169)
6. Yuan R, Zhu X, Radich JP, Ao P. 2016 From molecular interaction to acute promyelocytic leukemia: Calculating leukemogenesis and remission from endogenous molecular-cellular network. *Sci. Rep.* **6**, 24307. (doi:10.1038/srep24307)
7. Yuan R *et al.* 2016 Core level regulatory network of osteoblast as molecular mechanism for osteoporosis and treatment. *Oncotarget* **7**, 3692–3701. (doi:10.18632/oncotarget.6923)
8. Wang J, Yuan R, Zhu X, Ao P. 2020 Adaptive Landscape Shaped by Core Endogenous Network Coordinates Complex Early Progenitor Fate Commitments in Embryonic Pancreas. *Sci. Rep.* **10**, 1112. (doi:10.1038/s41598-020-57903-0)
9. Su H, Wang G, Yuan R, Wang J, Tang Y, Ao P, Zhu X. 2017 Decoding early myelopoiesis from dynamics of core endogenous network. *Sci. China Life Sci.* **60**, 627–646. (doi:10.1007/s11427-017-9059-y)
10. Kauffman SA. 1969 Metabolic stability and epigenesis in randomly constructed genetic nets. *J. Theor. Biol.* **22**, 437–467. (doi:10.1016/0022-5193(69)90015-0)

11. Kauffman SA. 1969 Homeostasis and Differentiation in Random Genetic Control Networks. *Nature* **224**, 177–178. (doi:10.1038/224177a0)
12. Monod J, Jacob F. 1961 General Conclusions: Teleonomic Mechanisms in Cellular Metabolism, Growth, and Differentiation. *Cold Spring Harb. Symp. Quant. Biol.* **26**, 389–401. (doi:10.1101/SQB.1961.026.01.048)
13. Long HK, Prescott SL, Wysocka J. 2016 Ever-Changing Landscapes: Transcriptional Enhancers in Development and Evolution. *Cell* **167**, 1170–1187. (doi:10.1016/j.cell.2016.09.018)
14. Enver T, Pera M, Peterson C, Andrews PW. 2009 Stem Cell States, Fates, and the Rules of Attraction. *Cell Stem Cell* **4**, 387–397. (doi:10.1016/j.stem.2009.04.011)
15. Huang S, Eichler G, Bar-Yam Y, Ingber DE. 2005 Cell Fates as High-Dimensional Attractor States of a Complex Gene Regulatory Network. *Phys. Rev. Lett.* **94**, 128701. (doi:10.1103/PhysRevLett.94.128701)
16. Huang S, Kauffman SA. 2012 *Complex Gene Regulatory Networks – from Structure to Biological Observables: Cell Fate Determination*. New York: Springer. (doi:https://doi.org/10.1007/978-1-4614-1800-9_35)
17. Huang S. 2012 The molecular and mathematical basis of Waddington’s epigenetic landscape: A framework for post-Darwinian biology? *BioEssays* **34**, 149–157. (doi:10.1002/bies.201100031)
18. Moris N, Pina C, Arias AM. 2016 Transition states and cell fate decisions in epigenetic landscapes. *Nat. Rev. Genet.* **17**, 693–703. (doi:10.1038/nrg.2016.98)
19. Zhu X-M, Yin L, Hood L, Ao P. 2004 Calculating biological behaviors of epigenetic states in the phage λ life cycle. *Funct. Integr. Genomics* **4**, 188–195. (doi:10.1007/s10142-003-0095-5)
20. Aldana M, Coppersmith S, Kadanoff LP. 2003 Boolean Dynamics with Random Couplings. In *Perspectives and Problems in Nonlinear Science*, pp. 23–89. New York, NY: Springer New York. (doi:10.1007/978-0-387-21789-5_2)